



# Contribution of meridional overturning circulation and sea ice changes to large-scale temperature asymmetries in CMIP6 overshoot scenarios

Pedro José Roldán-Gómez[1], Pablo Ortega[1], and Markus G. Donat[1,2]

[1]Barcelona Supercomputing Center, Earth Sciences Department, Barcelona, Spain
[2]Institució Catalana de Recerca i Estudis Avançats (ICREA), Barcelona, Spain

**Correspondence:** Pedro José Roldán-Gómez (pedro.roldan@bsc.es)

**Abstract.**

Analysis of overshoot scenarios, characterized by a peak in radiative forcing levels followed by a decline, show that changes during the $CO_2$ increasing phase are not necessarily compensated during the $CO_2$ decreasing phase, particularly at regional level. Even if the global mean temperature may recover after the overshoot, at the regional level the situation post-overshoot may differ from the situation pre-overshoot, with spatial patterns characterized by large-scale temperature asymmetries. These asymmetries, found between Northern (NH) and Southern Hemisphere (SH), between high and mid-latitudes of the NH, and between western and eastern areas of the Southern Ocean, alter atmospheric dynamics and through it the hydroclimate. Changes in the sea ice, changes in the ocean circulation and heat transport, and thermal inertia of the ocean have been identified as potential sources of hysteresis, highlighting the impact of oceanic changes in the behavior of atmospheric variables in case of overshoot. This work analyzes SSP5-3.4OS and SSP1-1.9 overshoot experiments from the Coupled Model Intercomparison Project Phase 6 (CMIP6) to assess how well these mechanisms can explain the large-scale temperature asymmetries that characterize the difference between pre-overshoot and post-overshoot states. These analyses show that the relative contribution of each mechanism strongly depends on the model. Certain models like MRI-ESM2-0 are mainly impacted by changes in the Atlantic Meridional Overturning Circulation (AMOC), others like CNRM-ESM2-1 show a relevant impact of sea ice changes in high-latitude areas, and others like IPSL-CM6A-LR show also a relevant impact of changes in the Southern Meridional Overturning Circulation (SMOC). Inter-model differences in the contributions of the meridional overturning can be associated with different climatologies of Mixed Layer Depth (MLD) in the northern North Atlantic (NNA) and in certain areas of the Southern Ocean. Despite these differences across models, all the mechanisms contribute to shape the regional temperatures after overshoot, with the temperature asymmetries between NH and SH mainly explained by changes in the AMOC, those between high and mid-latitudes of the NH by sea ice changes, and those between western and eastern areas of the Southern Ocean by the SMOC. These results highlight the importance of model intercomparison and analysis of ocean dynamics to understand the regional impacts of an overshoot, and more generally the responses to forcing changes.



## 1 Introduction

The increasing probability of exceeding the temperature targets of the Paris Agreement of 2015 (Raftery et al., 2017), as a result
of a delay of effective and consequent mitigation measures (IPCC, 2022), increases the likelihood of overshoot scenarios,
in which global average temperature surpasses the target of 1.5°C above pre-industrial levels (United Nations / Framework
Convention on Climate Change, 2015) and is recovered afterwards with net-negative emissions (Gasser et al., 2015). However,
large uncertainties exist for these possible scenarios, including the feasibility of recovering the pre-overshoot temperatures
with large-scale Carbon Dioxide Removal (CDR) in the expected timescales, and the long-term climate risks associated with
reinforced Earth-system feedbacks characteristic of such scenarios (Schleussner et al., 2024).

Considering the increasing interest in these overshoots, the Coupled Model Intercomparison Project Phase 6 (CMIP6; Eyring
et al., 2016) included in the ScenarioMIP (O'Neill et al., 2016) two scenarios with forcing pathways that reach a peak before
starting a forcing decline: SSP5-3.4OS and SSP1-1.9. These two scenarios present an overshoot, but at different forcing levels.
While SSP1-1.9 includes mitigation actions to meet the 1.5°C target of the Paris Agreement with a moderate overshoot, SSP5-
3.4OS follows the unmitigated scenario SSP5-8.5 up to 2040 and starts an aggressive mitigation afterwards (Tebaldi et al.,
2021).

The analysis of these scenarios shows that even if global temperatures revert, the impact on regional temperature, precip-
itation and climate extremes may remain for decades (Pfleiderer et al., 2024). This regional irreversibility, understood as a
post-overshoot state different from the pre-overshoot state with the same $CO_2$ concentration levels and with the same global
temperature, has been associated with hemispherical temperature asymmetries between pre-overshoot and post-overshoot states
(Roldán-Gómez et al., 2025), impacting the precipitation of tropical areas through changes in the position of the Intertropical
Convergence Zone (ITCZ). These temperature asymmetries may be linked to persistent changes in the Ocean Heat Transport
(OHT) and to different thermal inertia depending on the region (Roldán-Gómez et al., 2025), but also to other factors like
anthropogenic aerosol emissions (England et al., 2021) and ice melting (Li et al., 2020).
The analysis of the SSP1-1.9 and SSP5-3.4OS scenarios confirms the relevant role of hysteresis mechanisms in shaping
regional temperature and precipitation after global temperature overshoots, understood as the dependence of the climate system
not only on the current $CO_2$ concentration but on the $CO_2$ pathway. Relevant hysteresis mechanisms have been found on the
large-scale hydrology, with persistent changes in global precipitation after a $CO_2$ overshoot (Cao et al., 2011), with changes in
the position of the ITCZ associated with a delayed energy exchange between the tropics and extratropics (Kug et al., 2022),
and with impacts on the monsoon system, with a contrasting eastern-western hemispheric monsoon response in the NH (Oh
et al., 2022) and an enhanced El Niño-like warming pattern altering the East Asian summer monsoon (Song et al., 2022).

Hysteresis is also found in ocean dynamics and sea level changes (Palter et al., 2018), in Antarctic ice sheet retreat (Garbe
et al., 2020; Bochow et al., 2023), in ocean acidification (Jiang et al., 2024) and in ocean carbon cycle feedbacks (Schwinger and
Tjiputra, 2018) . Li et al. (2024) show that the post-overshoot state in SSP5-3.4OS is characterized by a persistent weakening of
Atlantic Meridional Overturning Circulation (AMOC), with impacts on the OHT also for the Southern Ocean and, to a lesser
extent, for the Indo-Pacific basin. However, there are large uncertainties in these changes, with responses of the AMOC to





forcing changes that strongly depend on the model considered (Sgubin et al., 2017). In global warming conditions, a decrease of OHT is generally found (Mecking and Drijfhout, 2023), both in the NH and in the SH, but models disagree on the presence of tipping points, with only some models showing a local convection collapse in the subpolar North Atlantic, and only a few

of them presenting a full AMOC disruption (Sgubin et al., 2017). A collapse of the AMOC may have important impacts at a global scale (Orihuela-Pinto et al., 2022), stressing the need to understand in which conditions it may occur (Jackson et al., 2023). Baker et al. (2025) show that the collapse of the AMOC is unlikely even under extreme forcing conditions, and may only occur in case a strong Pacific Meridional Overturning Circulation (PMOC) emerges.

  This work analyzes overshoot scenarios from CMIP6 (SSP5-3.4OS and SSP1-1.9) to investigate how irreversibility mecha-

nisms associated with ocean dynamics and sea ice changes contribute to generate large-scale temperature asymmetries in the post-overshoot climate, with a particular focus on how different models characterize the relative contribution of each mechanism. Asymmetries between Northern (NH) and Southern Hemisphere (SH), between high and mid-latitudes of the NH, and between western and eastern areas of the Southern Ocean are correlated with changes in the OHT, including contributions of the AMOC in the Atlantic basin and of the Southern Meridional Overturning Circulation (SMOC) in the Southern Ocean, as

well as with changes in the ice cover in both the NH and SH. These analyses, covering the entire range of responses present in CMIP6 models, allow for a complete assessment of the relative contribution of each mechanism to temperature asymmetries, and for a better understanding of the different responses found in regional climates after the overshoot.

## 2 Methods

The analyses have been based on the simulations of CMIP6 overshoot experiments SSP5-3.4OS and SSP1-1.9 listed in Table 1.

This includes 165 simulations from 11 different models for SSP1-1.9, all of them covering the period from 1850 to 2100, and 16 simulations from eight models for SSP5-3.4OS, with four of them covering an extended period from 1850 to 2300. As shown in Table 1, ensembles of simulations with the same forcing specifications and different initial conditions are available for some of the models, including CanESM5 and UKESM1-0-LL for SSP5-3.4OS, and CanESM5, MIROC6, MIROC-ES2L, IPSL-CM6A-LR, EC-Earth3, MPI-ESM1-2-LR, MRI-ESM2-0, and UKESM1-0-LL for SSP1-1.9. The use of ensemble averages

allows to better separate the responses to forcing changes from potential contributions of internal variability. In addition to the ensemble of simulations from each individual model, combined ensembles have been also considered, including the ensemble of all simulations (ALL) for SSP1-1.9 and SSP5-3.4OS, and the ensemble of simulations covering the extended period (EXT) for SSP5-3.4OS. The ensemble averages of the ALL and EXT ensembles have been computed by averaging all the simulations from each model to obtain a per-model average in a first step and by averaging all the models in a second step, so that all the

models contribute with the same weight to the multi-model average.

  As shown in Fig. 1, the resolution of the atmospheric component of the selected models varies from 2.8º to 0.7º, while the resolution of the ocean component varies from 1.7º to 0.5º. To allow for combined analyses, a longitude-latitude grid has been considered both for the atmospheric and the ocean variables, and all the simulations have been remapped with a bilinear interpolation to a common grid resolution of 2.8º, the coarsest among the analyzed climate models.





**Table 1.** Climate models analyzed, available simulations for the SSP5-3.4OS and SSP1-1.9 experiments considered in this work, resolution of atmospheric (ResA) and ocean component (ResO) of each model and associated references. For the SSP5-3.4OS, the number of simulations covering the extended period (up to 2300) is also included in the column (EXT).

| Model | SSP5-3.4OS | (EXT) | SSP1-1.9 | ResA | ResO | References |
|---|---|---|---|---|---|---|
| ACCESS-CM2 | 1 (r1i1p1f1) | 0 | 0 | 1.9º x 1.2º | 1.0º x 1.2º | Ziehn et al. (2021) |
| CanESM5 | 5 (r[1-5]i1p1f1) | 1 (r1i1p1f1) | 50 (r[1-25]i1p[1-2]f1) | 2.8º x 2.8º | 1.0º x 1.2º | Swart et al. (2019a, b) |
| CMCC-ESM2 | 1 (r1i1p1f1) | 0 | 0 | 1.3º x 0.9º | 1.0º x 1.2º | Lovato et al. (2021) |
| CNRM-ESM2-1 | 1 (r1i1p1f2) | 1 (r1i1p1f2) | 1 (r1i1p1f2) | 1.4º x 1.4º | 1.0º x 1.2º | Voldoire (2019a, b) |
| EC-Earth3 | 0 | 0 | 6 (r[1-4]i1p1f1) | 0.7º x 0.7º | 1.0º x 1.2º | Döscher et al. (2022); EC-Earth-Consortium (2019a, b, c) |
| FGOALS-g3 | 1 (r1i1p1f1) | 0 | 1 (r1i1p1f1) | 2.0º x 2.3º | 1.0º x 1.7º | Li (2019, 2020) |
| GFDL-ESM4 | 0 | 0 | 1 (r1i1p1f1) | 1.3º x 1.0º | 0.5º x 0.7º | John et al. (2018) |
| IPSL-CM6A-LR | 1 (r1i1p1f1) | 1 (r1i1p1f1) | 6 (r[1-4,6,14]i1p1f1) | 2.5º x 1.3º | 1.0º x 1.1º | Boucher et al. (2019a, b) |
| MIROC6 | 0 | 0 | 50 (r[1-50]i1p1f1) | 1.4º x 1.4º | 1.0º x 1.4º | Shiogama et al. (2019) |
| MIROC-ES2L | 0 | 0 | 10 (r[1-10]i1p1f2) | 2.8º x 2.8º | 1.0º x 1.4º | Tachiiri et al. (2019) |
| MPI-ESM1-2-LR | 0 | 0 | 30 (r[1-30]i1p1f1) | 1.9º x 1.9º | 1.4º x 1.6º | Schupfner et al. (2021) |
| MRI-ESM2-0 | 1 (r1i1p1f1) | 1 (r1i1p1f1) | 5 (r[1-5]i1p1f1) | 1.1º x 1.1º | 1.0º x 1.0º | Yukimoto et al. (2019a, b) |
| UKESM1-0-LL | 5 (r[1-4,8]i1p1f2) | 0 | 5 (r[1-4,8]i1p1f2) | 1.9º x 1.3º | 1.0º x 1.1º | Good et al. (2019a, b) |

In a first step, analyses have been focused on annual Sea Surface Temperatures (SST), to characterize the temperature asymmetries generated during the overshoot. To focus on the long-term variability, analyses have been based on temporal evolutions filtered with a 10 year moving average and comparisons of 20 year pre- and post-overshoot periods. The post-overshoot state has been compared with the pre-overshoot state with the same global average of surface air temperature, which is reached in 2034 for SSP5-3.4OS and in 2030 for SSP1-1.9, and the pre-overshoot state with the same $CO_2$ concentration

(Meinshausen et al., 2020), reached in 2015 both for SSP5-3.4OS and SSP1-1.9. Considering these dates and in order to use a reference period large enough to focus on the long-term variability, our pre-overshoot reference for all analyses (except those stated otherwise) is the period from 2020 to 2039. Despite their secondary role with respect to changes in $CO_2$ concentration, potential differences in aerosol emissions between SSP5-3.4OS and SSP1-1.9 have been assessed by considering the aerosol emissions provided by Feng et al. (2020).

To evaluate temperature asymmetries between pre- and post-overshoot states, and considering the results from Roldán-Gómez et al. (2025), the regions included in Fig. 1 have been considered. In particular, the asymmetry between NH and SH has been characterized as the difference between regional averages of SST for the extratropical ocean areas of the NH (EN) and the



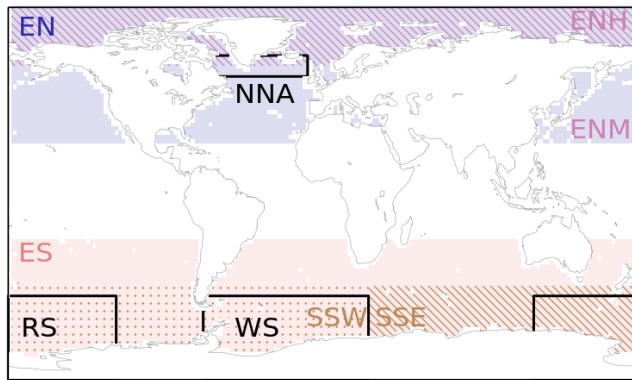

**Figure 1.** Regions considered for the analysis of temperature asymmetries and MLD, including extratropical ocean areas of the NH (EN; 23° N - 90° N), high-latitude extratropical ocean areas of the NH (ENH; 60° N - 90° N), mid-latitude extratropical ocean areas of the NH (ENM; 23° N - 60° N), extratropical ocean areas of the SH (ES; 90° S - 23° S), south-western Southern Ocean (SSW; 90° S - 45° S; 180° W - 25° E), south-eastern Southern Ocean (SSE; 90° S - 45° S; 25° E - 180° E), northern North Atlantic (NNA; 55° N - 65° N; 70° W - 10° W), Weddell Sea (WS; 90° S - 50° S; 70° W - 25° E), and Ross Sea (RS; 90° S - 50° S; 120° E - 120° W).

extratropical ocean areas of the SH (ES), the asymmetry between medium and high latitudes of the NH has been characterized with the difference between regional averages for mid-latitude extratropical ocean areas of the NH (ENM) and high-latitude

extratropical ocean areas of the NH (ENH), and the asymmetry between western and eastern areas of the Southern Ocean has been characterized with the difference between regional averages for south-western Southern Ocean (SSW) and south-eastern Southern Ocean (SSE). To confirm that the results are not sensitive to the particular definition of these regions, results for EN, ES and ENM but including only the Atlantic basin (ENATL, ESATL and ENMATL) are included in Appendix A.

    In a second step, some mechanisms and processes potentially contributing to these temperature asymmetries have been

explored, associated with the global, Atlantic and Pacific OHT, the Atlantic mass transport, the sea ice concentration, and the Mixed Layer Depth (MLD) provided by the models. A set of indices have been defined to evaluate their variability, including the Atlantic OHT at 26ºN and 30ºS, the global OHT at 50ºS, the total sea ice area for the NH and the SH, the AMOC index (computed as the maximum Atlantic mass transport between 30°S and 60°N and between 500 m and the bottom), the average March MLD for the northern North Atlantic (NNA, as defined in Fig. 1), and the average September MLD for Weddell Sea and

Ross Sea (WS and RS, as defined in Fig. 1). The difference between pre- and post-overshoot states and the temporal evolution of these indices have been evaluated for each simulation of SSP5-3.4OS and SSP1-1.9 experiments, as well as for the ensemble average of each individual model and the average of ALL and EXT ensembles.



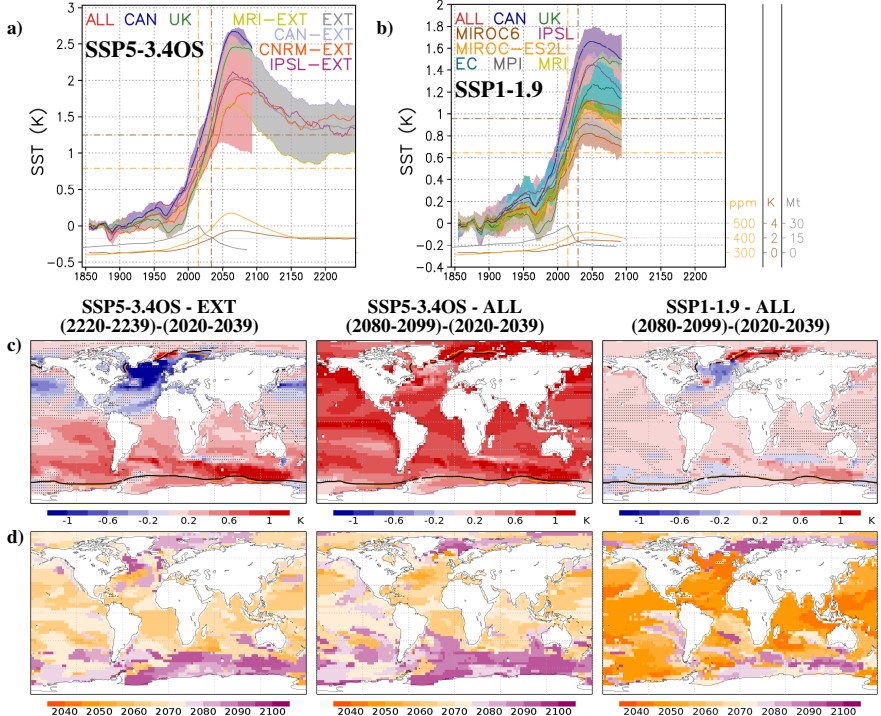

**Figure 2. (a,b)** Global average of Sea Surface Temperature (SST) anomaly with respect to 1861-1880 obtained from the CMIP6 simulations of experiments **(a)** SSP5-3.4OS and **(b)** SSP1-1.9, including the ensemble average of all the models (ALL), as well as the average of the ensembles from CanESM5 (CAN), UKESM1-0-LL (UK), MIROC6, MIROC-ES2L, IPSL-CM6A-LR (IPSL), EC-Earth3 (EC), MPI-ESM1-2-LR (MPI), and MRI-ESM2-0 (MRI), and, for SSP5-3.4OS, the ensemble average of simulations extended up to 2300 (EXT) and the individual extended simulations from CanESM5 (CAN-EXT), CNRM-ESM2-1 (CNRM-EXT), IPSL-CM6A-LR (IPSL-EXT), and MRI-ESM2-0 (MRI-EXT). The spread of individual simulations within each ensemble is included with a shading. Yellow, gray and brown curves in the lower part of each panel respectively show the $CO_2$ concentration from Meinshausen et al. (2020), the anthropogenic aerosol emissions from Feng et al. (2020), and the global surface air temperature obtained with the SSP1-1.9 ALL ensemble and the SSP5-3.4OS EXT ensemble. The vertical lines show the year before the overshoot with the same $CO_2$ concentration (yellow line) and global surface air temperature (brown line) as at the end of the run (2100 for the SSP1-1.9 ALL ensemble and 2300 for the SSP5-3.4OS EXT ensemble), while the horizontal lines represent the value of SST in the SSP1-1.9 ALL ensemble and in the SSP5-3.4OS EXT ensemble for those years. **(c)** Difference between the ensemble mean, temporal average values of SST for **(left)** the periods 2220-2239 and 2020-2039 obtained with the SSP5-3.4OS EXT ensemble, **(center)** the periods 2080-2099 and 2020-2039 obtained with the SSP5-3.4OS ALL ensemble, and **(right)** the periods 2080-2099 and 2020-2039 obtained with the SSP1-1.9 ALL ensemble. Contours of 15% of sea ice concentration are included in the maps, both for the periods 2220-2239 and 2080-2099 (yellow) and for the reference period 2020-2039 (black). Stippling indicates locations where the differences are not significant (t-test with p<0.05). **(d)** Year of maximum SST for **(left)** the SSP5-3.4OS EXT ensemble, **(center)** the SSP5-3.4OS ALL ensemble, and **(right)** the SSP1-1.9 ALL ensemble. Stippling indicates locations where the overshoot is associated with a temperature minimum. In those cases, the year of the minimum is considered instead.





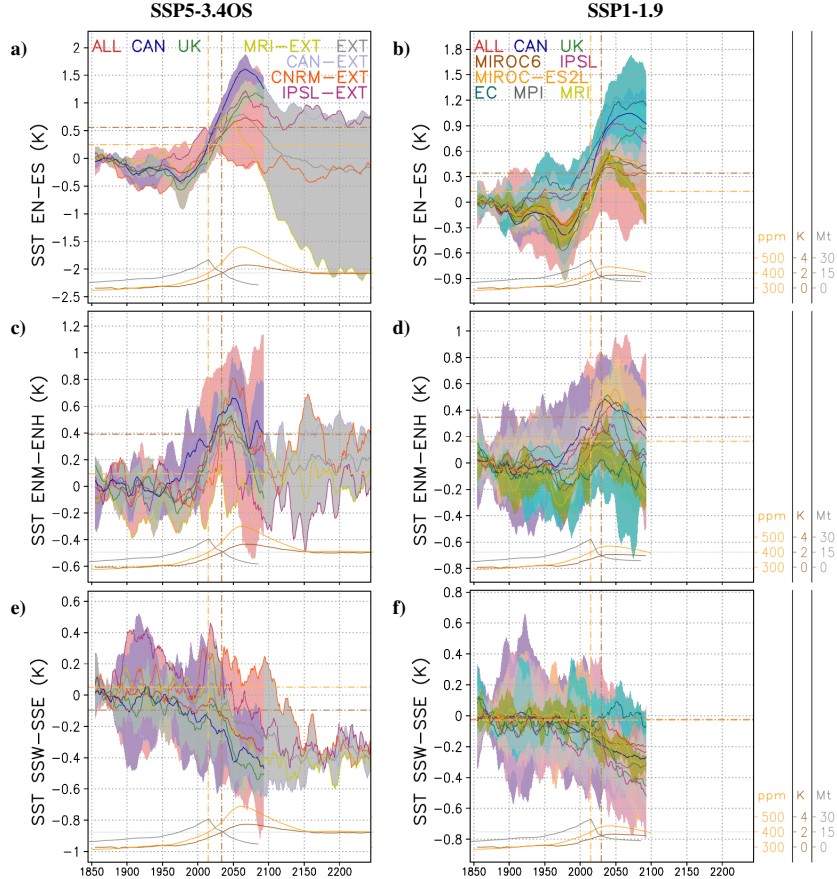

**Figure 3. (a,b)** Anomaly of Sea Surface Temperature (SST) difference between the extratropical ocean areas of the NH (EN) and the extratropical ocean areas of the SH (ES) with respect to 1861-1880, obtained from the CMIP6 simulations of experiments **(a)** SSP5-3.4OS and **(b)** SSP1-1.9, including the ensemble average of all the models (ALL), as well as the average of the ensembles from CanESM5 (CAN), UKESM1-0-LL (UK), MIROC6, MIROC-ES2L, IPSL-CM6A-LR (IPSL), EC-Earth3 (EC), MPI-ESM1-2-LR (MPI), and MRI-ESM2-0 (MRI), and, for SSP5-3.4OS, the ensemble average of simulations extended up to 2300 (EXT) and the individual extended simulations from CanESM5 (CAN-EXT), CNRM-ESM2-1 (CNRM-EXT), IPSL-CM6A-LR (IPSL-EXT), and MRI-ESM2-0 (MRI-EXT). The spread of individual simulations within each ensemble is included with a shading. Yellow, gray and brown curves in the lower part of each panel respectively show the $CO_2$ concentration from Meinshausen et al. (2020), the anthropogenic aerosol emissions from Feng et al. (2020), and the global surface air temperature obtained with the SSP1-1.9 ALL ensemble and the SSP5-3.4OS EXT ensemble. The vertical lines show the year before the overshoot with the same $CO_2$ concentration (yellow line) and global surface air temperature (brown line) as at the end of the run (2100 for the SSP1-1.9 ALL ensemble and 2300 for the SSP5-3.4OS EXT ensemble), while the horizontal lines represent the value of SST difference in the SSP1-1.9 ALL ensemble and in the SSP5-3.4OS EXT ensemble for those years. **(c,d)** Same as Fig. 3a,b, but for the mid-latitude extratropical ocean areas of the NH (ENM) and the high-latitude extratropical ocean areas of the NH (ENH). **(e,f)** Same as Fig. 3a,b, but for the south-western Southern Ocean (SSW) and the south-eastern Southern Ocean (SSE).



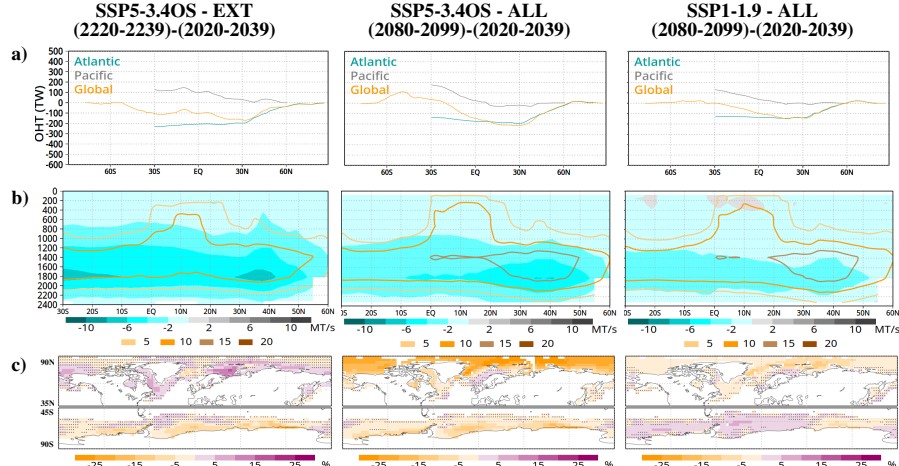

**Figure 4.** Difference between the ensemble mean, temporal average values of **(a)** Ocean Heat Transport (OHT; positive to the North), **(b)** Atlantic mass transport (positive to the South), and **(c)** sea ice concentration for **(left)** the periods 2220-2239 and 2020-2039 obtained with the SSP5-3.4OS EXT ensemble, **(center)** the periods 2080-2099 and 2020-2039 obtained with the SSP5-3.4OS ALL ensemble, and **(right)** the periods 2080-2099 and 2020-2039 obtained with the SSP1-1.9 ALL ensemble. Differences of OHT are shown per latitude, for the Atlantic basin, the Pacific basin, and the global zonal average. Differences of Atlantic mass transport are shown per latitude and level. Contours in (b) indicate climatological values of Atlantic mass transport obtained during the period 1861-1880. Stippling in (c) indicates locations where the differences are not significant (t-test with $p<0.05$).

## 3 Results

### 3.1 Temperature asymmetries

The evolution of the global average of SST for the SSP5-3.4OS and SSP1-1.9 experiments is shown in Fig. 2a and Fig. 2b. After the stabilization, the global average of SST for the SSP5-3.4OS EXT ensemble (gray line in Fig. 2a) and the SSP1-1.9 ALL ensemble (red line in Fig. 2b) is larger than the expected value considering the situation before the overshoot with the same global average of surface air temperature (horizontal brown line) or the situation before the overshoot with the same $CO_2$ concentration (horizontal yellow line). This shows that the heat accumulated by the ocean during the $CO_2$ increasing phase

has not been totally released to the atmosphere afterwards, in line with the results from Roldán-Gómez et al. (2025). Despite this common result for the EXT and ALL ensembles, strong differences exist across the simulations and models. For example, for the extended simulation of MRI-ESM2-0 in SSP5-3.4OS and for the ensemble average of MIROC6, MIROC-ES2L, MPI-ESM1-2-LR, and MRI-ESM2-0 in SSP1-1.9, the global average of SST after overshoot is lower than that of before.

Regarding the spatial patterns, the comparison of post-overshoot (2220-2239) and pre-overshoot (2020-2039) mean states

for the average of SSP5-3.4OS EXT ensemble (Fig. 2c, left) highlights an asymmetry between extratropical areas of the NH (region EN from Fig. 1), with colder values after overshoot, and extratropical areas of the SH (region ES from Fig. 1), with warmer values after overshoot. A more localized asymmetry is also found between mid-latitude areas of the NH (region ENM





from Fig. 1), mainly presenting colder post-overshoot states, and high-latitude areas of the NH (region ENH from Fig. 1), which present warmer values. This asymmetry between ENM and ENH is more important when comparing the post-overshoot

(2080-2099) and pre-overshoot (2020-2039) states of SSP1-1.9 ALL ensemble (Fig. 2c, right). In this case, also an asymmetry between eastern (region SSE from Fig. 1) and western Southern Ocean (region SSW from Fig. 1) is found, being SSE mainly warmer and SSW mainly colder after the overshoot.

This different behavior after overshoot depending on the region may be associated with a different timing of the temperature maximum (Roldán-Gómez et al., 2025). Figure 2d shows the year of the maximum for the EXT and ALL ensembles of SSP5-

3.4OS and for the ALL ensemble of SSP1-1.9. The figure shows that most tropical areas and mid-latitude extratropical areas of the NH reach the temperature maximum before 2070 for SSP5-3.4OS and before 2050 for SSP1-1.9, while for most areas of the Southern Ocean the temperature maximum is delayed, between 2080 and 2100 for SSP5-3.4OS and between 2060 and 2090 for SSP1-1.9. The spatial patterns depicting the year of the maximum in SST (Fig. 2d) show similar features to the spatial patterns of post- and pre-overshoot difference (Fig. 2c), suggesting a link between the timing of the local maximum and the

state after overshoot. Areas with delayed maximum, like most areas of the Southern Ocean, can be associated with a larger warming during the $CO_2$ increasing phase which is not fully compensated during the $CO_2$ decreasing phase. Conversely, areas with an early maximum, like mid-latitude extratropical areas of the NH, can be associated with less warming during the $CO_2$ increasing phase and an overcompensation during the $CO_2$ decreasing phase.

The comparison of different models (Fig. 3) for indices that represent the aforementioned asymmetries reveals that these

asymmetries are not equally present in all models. For example, the simulations from MRI-ESM2-0 show a strong EN-ES asymmetry, both for SSP5-3.4OS (Fig. 3a) and SSP1-1.9 (Fig. 3b), while for other simulations like those from IPSL-CM6A-LR this asymmetry is not clearly found. Something similar happens for the ENM-ENH asymmetry (Fig. 3c,d). Even if the averages of the SSP5-3.4OS EXT ensemble (gray line in Fig. 3c) and the SSP1-1.9 ALL ensemble (red line in Fig. 3d) show more negative values of ENM-ENH temperature after overshoot, this is only found for some simulations like those from

MRI-ESM2-0 and IPSL-CM6A-LR, while some others like those from CNRM-ESM2-1 or MIROC-ES2L do not show a clear asymmetry. Less discrepancies exist for the SSW-SSE asymmetry (Fig. 3e,f), for which most model simulations show colder temperatures in the western sector after the overshoot, even if some particular simulations like those from EC-Earth3 show comparable temperatures in both regions (Fig. 3f).

## 3.2 Meridional overturning circulation and sea ice changes

The results from Fig. 2 and Fig. 3 show persistent changes in the regional temperatures after the overshoot, which can be associated with a different timing of the temperature maximum, but these results strongly depend on the model. To better understand the reasons behind this behavior, different mechanisms potentially contributing to regional hysteresis in case of overshoot have been analyzed, including changes in the meridional overturning circulation (Li et al., 2024) and changes in the sea ice coverage (Li et al., 2020).

Figure 4 shows the difference between post-overshoot (2220-2239 for SSP5-3.4OS EXT ensemble and 2080-2099 for SSP5-3.4OS and SSP1-1.9 ALL ensemble) and pre-overshoot (2020-2039) states for the global, Atlantic and Pacific OHT (Fig. 4a)



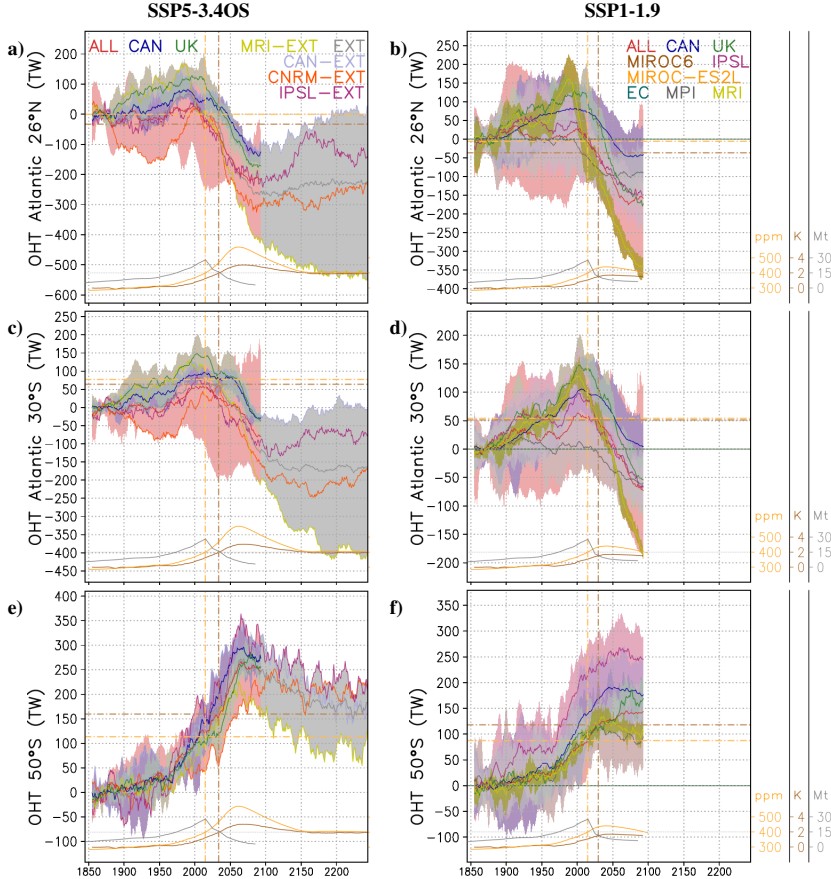

**Figure 5. (a,b)** Anomaly of Atlantic Ocean Heat Transport (OHT; positive to the North) at 26°N with respect to 1861-1880 obtained from the CMIP6 simulations of experiments **(a)** SSP5-3.4OS and **(b)** SSP1-1.9, including the ensemble average of all the models (ALL), as well as the average of the ensembles from CanESM5 (CAN), UKESM1-0-LL (UK), MIROC6, MIROC-ES2L, IPSL-CM6A-LR (IPSL), EC-Earth3 (EC), MPI-ESM1-2-LR (MPI), and MRI-ESM2-0 (MRI), and, for SSP5-3.4OS, the ensemble average of simulations extended up to 2300 (EXT) and the individual extended simulations from CanESM5 (CAN-EXT), CNRM-ESM2-1 (CNRM-EXT), IPSL-CM6A-LR (IPSL-EXT), and MRI-ESM2-0 (MRI-EXT). The spread of individual simulations within each ensemble is included with a shading. Yellow, gray and brown curves in the lower part of each panel respectively show the $CO_2$ concentration from Meinshausen et al. (2020), the anthropogenic aerosol emissions from Feng et al. (2020), and the global surface air temperature obtained with the SSP1-1.9 ALL ensemble and the SSP5-3.4OS EXT ensemble. The vertical lines show the year before the overshoot with the same $CO_2$ concentration (yellow line) and global surface air temperature (brown line) as at the end of the run (2100 for the SSP1-1.9 ALL ensemble and 2300 for the SSP5-3.4OS EXT ensemble), while the horizontal lines represent the value of Atlantic OHT at 26°N in the SSP1-1.9 ALL ensemble and in the SSP5-3.4OS EXT ensemble for those years. **(c,d)** Same as Fig. 5a,b, but for the Atlantic OHT at 30°S. **(e,f)** Same as Fig. 5a,b, but for the global OHT at 50°S.





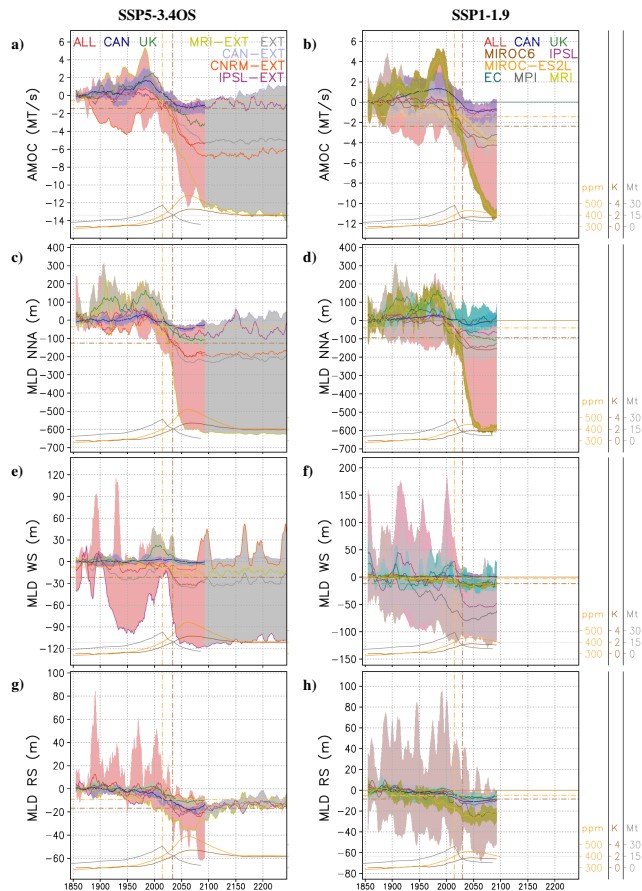

**Figure 6. (a,b)** Anomaly of Atlantic Meridional Overturning Circulation (AMOC) index with respect to 1861-1880 obtained from the CMIP6 simulations of experiments **(a)** SSP5-3.4OS and **(b)** SSP1-1.9, including the ensemble average of all the models (ALL), as well as the average of the ensembles from CanESM5 (CAN), UKESM1-0-LL (UK), MIROC6, MIROC-ES2L, IPSL-CM6A-LR (IPSL), EC-Earth3 (EC), MPI-ESM1-2-LR (MPI), and MRI-ESM2-0 (MRI), and, for SSP5-3.4OS, the ensemble average of simulations extended up to 2300 (EXT) and the individual extended simulations from CanESM5 (CAN-EXT), CNRM-ESM2-1 (CNRM-EXT), IPSL-CM6A-LR (IPSL-EXT), and MRI-ESM2-0 (MRI-EXT). The spread of individual simulations within each ensemble is included with a shading. Yellow, gray and brown curves in the lower part of each panel respectively show the $CO_2$ concentration from Meinshausen et al. (2020), the anthropogenic aerosol emissions from Feng et al. (2020), and the global surface air temperature obtained with the SSP1-1.9 ALL ensemble and the SSP5-3.4OS EXT ensemble. The vertical lines show the year before the overshoot with the same $CO_2$ concentration (yellow line) and global surface air temperature (brown line) as at the end of the run (2100 for the SSP1-1.9 ALL ensemble and 2300 for the SSP5-3.4OS EXT ensemble), while the horizontal lines represent the value of AMOC index in the SSP1-1.9 ALL ensemble and in the SSP5-3.4OS EXT ensemble for those years. **(c,d)** Same as Fig. 6a,b, but for the March Mixed Layer Depth (MLD) in the northern North Atlantic (NNA) region. **(e,f)** Same as Fig. 6a,b, but for the September MLD in the Weddell Sea (WS) region. **(g,h)** Same as Fig. 6a,b, but for the September MLD in the Ross Sea (RS) region.





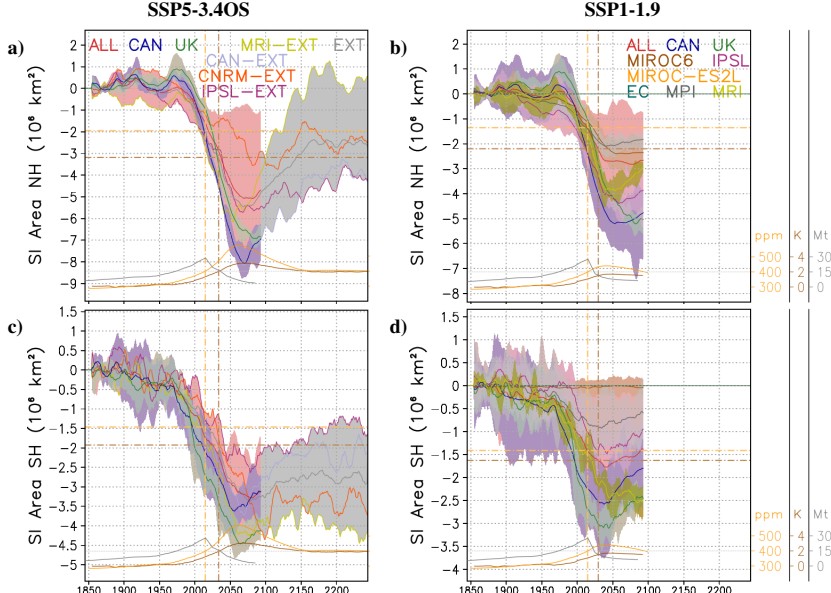

**Figure 7. (a,b)** Anomaly of Sea Ice (SI) area in the NH with respect to 1861-1880 obtained from the CMIP6 simulations of experiments **(a)** SSP5-3.4OS and **(b)** SSP1-1.9, including the ensemble average of all the models (ALL), as well as the average of the ensembles from CanESM5 (CAN), UKESM1-0-LL (UK), MIROC6, MIROC-ES2L, IPSL-CM6A-LR (IPSL), EC-Earth3 (EC), MPI-ESM1-2-LR (MPI), and MRI-ESM2-0 (MRI), and, for SSP5-3.4OS, the ensemble average of simulations extended up to 2300 (EXT) and the individual extended simulations from CanESM5 (CAN-EXT), CNRM-ESM2-1 (CNRM-EXT), IPSL-CM6A-LR (IPSL-EXT), and MRI-ESM2-0 (MRI-EXT). The spread of individual simulations within each ensemble is included with a shading. Yellow, gray and brown curves in the lower part of each panel respectively show the $CO_2$ concentration from Meinshausen et al. (2020), the anthropogenic aerosol emissions from Feng et al. (2020), and the global surface air temperature obtained with the SSP1-1.9 ALL ensemble and the SSP5-3.4OS EXT ensemble. The vertical lines show the year before the overshoot with the same $CO_2$ concentration (yellow line) and global surface air temperature (brown line) as at the end of the run (2100 for the SSP1-1.9 ALL ensemble and 2300 for the SSP5-3.4OS EXT ensemble), while the horizontal lines represent the value of SI area in the NH in the SSP1-1.9 ALL ensemble and in the SSP5-3.4OS EXT ensemble for those years. **(c,d)** Same as Fig. 7a,b, but for the SI area in the SH.

and the Atlantic mass transport (Fig. 4b). In the multi-model average, the post-overshoot state is characterized by a lower northward OHT in the Atlantic basin (Fig. 4a), particularly between 30ºS and 30ºN, associated with a weakened meridional mass transport (Fig. 4b). Changes in the Atlantic ocean mass transport are particularly strong where the overturning streamfunction

attains its climatological maximum, that is between 1200 and 2000 m and between 30ºN and 40ºN. In line with the analysis from Li et al. (2024) and Baker et al. (2025), the reduced heat transport in the Atlantic basin is partly compensated by increased northward OHT in the Pacific basin, between 30ºS and the equator (Fig. 4a). In the Southern Ocean, higher values of northward OHT are found on average between 60ºS and 30ºS for the SSP1-1.9 ALL ensemble (Fig. 4a, right), but this is not the case for the SSP5-3.4OS EXT ensemble (Fig. 4a, left). As for the case of temperatures, the temporal evolution of OHT for individual





models (Fig. 5) highlights a strong inter-model dispersion. Some models like MRI-ESM2-0 show a strong decrease of Atlantic OHT, both at 26ºN (Fig. 5a,b) and at 30ºS (Fig. 5c,d), while some others like CanESM5 and IPSL-CM6A-LR show a more limited decrease. Likewise, MRI-ESM2-0 shows a moderate strengthening of the northward OHT in the Southern Ocean (Fig. 5e,f), while some other models like CNRM-ESM2-1 and IPSL-CM6A-LR show stronger changes.

These changes in the OHT can be associated with changes in the AMOC (Fig. 6a,b). The AMOC index after overshoot is lower than before for most simulations, with the changes being particularly strong in the MRI-ESM2-0 simulations. The MLD in the northern North Atlantic region (NNA; Fig. 6c,d), which can be regarded as a driver of the AMOC through its major role in deep water formation, also shows important differences between pre- and post-overshoot states, particularly important for MRI-ESM2-0. In the Southern Ocean, the MLD in key regions like the Weddell and the Ross Seas (WS and RS, respectively; Fig. 6e,f and Fig. 6g,h) is particularly weakened in the IPSL-CM6A-LR simulations, consistent with the important changes found in the OHT at 50ºS for this model (Fig. 5e,f).

Regarding the sea ice concentration (Fig. 4c), the post-overshoot state for the SSP5-3.4OS EXT ensemble average is characterized by higher concentrations in the NH and lower concentrations in the SH (Fig. 4c, left), a feature that can be linked to the reduced global northward heat transport. By contrast, for the SSP1-1.9 ALL ensemble lower concentrations are found in the eastern Southern Ocean and the NH and higher concentrations are found in the western Southern Ocean (Fig. 4c, right). These differences between SSP5-3.4OS and SSP1-1.9 can be explained by the different models represented in each ensemble, as individual models show highly heterogeneous sea ice responses. As shown in Fig. 7a,c, the average of the SSP5-3.4OS EXT ensemble (gray line) shows a larger sea ice area in the NH and a smaller sea ice area in the SH after overshoot, but this mean response is dominated by the simulations from MRI-ESM2-0 and, to a lesser extent, CNRM-ESM2-1. For the average of SSP1-1.9 ALL (red line in Fig. 7b,d), the behavior is the opposite, with smaller sea ice area in the NH and larger sea ice area in the SH after overshoot.

### 3.3 Differences across models

Considering the large discrepancies across the models in the simulation of temperature changes (Fig. 3), the OHT and AMOC responses (Fig. 5 and 6), and the concomitant sea ice changes (Fig. 7), a per-model analysis is deemed necessary.

Figure 8 shows how the EN-ES temperature asymmetry —as characterized by the difference between the post-overshoot (2220-2239 and 2080-2099) and pre-overshoot (2020-2039) states —in individual simulations and their ensemble mean relate to analogous differences in a selection of indices potentially contributing to this asymmetry (i.e. Atlantic OHT at 26ºN, AMOC index, and MLD in NNA). The models with the most negative differences in the northward Atlantic OHT, like MRI-ESM2-0 and CNRM-ESM2-1, are also those showing the coldest temperatures in EN with respect to ES. The $R^2$ coefficient between northward Atlantic OHT at 26ºN and EN-ES asymmetry reaches 0.97 for SSP5-3.4OS EXT and 0.57 for SSP5-3.4OS ALL, both significant at a 95% confidence level (Fig. 8e). Even if the high correlation for SSP5-3.4OS EXT may be explained by the limited number of simulations covering the period up to 2300 (only 4 simulations), these results show a clear relationship between Atlantic OHT and the hemispheric temperature asymmetries. For the case of SSP1-1.9, MRI-ESM2-0 and CNRM-ESM2-1 are also the models showing the strongest OHT reduction and EN-ES asymmetry, even if the ensemble averages do



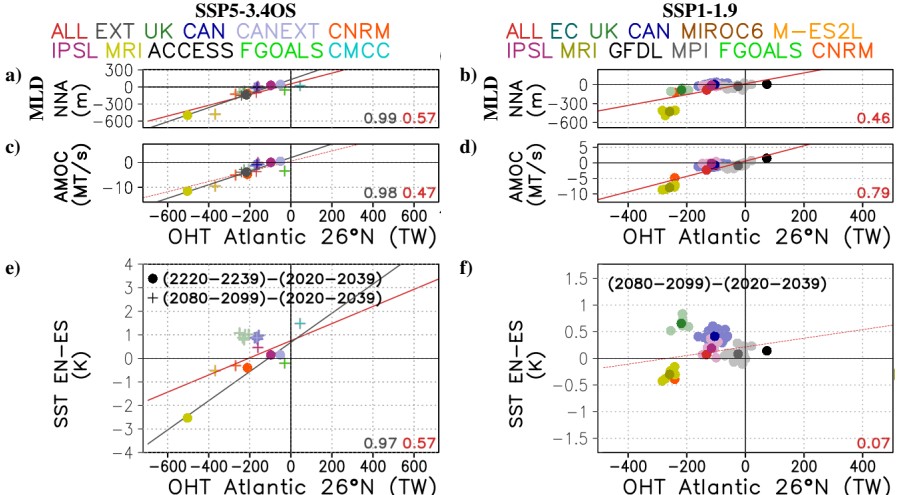

**Figure 8. (a,b)** March Mixed Layer Depth (MLD) in the northern North Atlantic (NNA) versus Atlantic Ocean Heat Transport (OHT) at 26ºN for the periods 2220-2239 and 2080-2099 with respect to the reference period 2020-2039, obtained with each simulation of **(a)** SSP5-3.4OS and **(b)** SSP1-1.9, as well as with the ensemble average of all the models (ALL), the ensemble average of simulations extended up to 2300 (EXT), and the ensemble average of each model containing several simulations. Regression lines and coefficients of determination ($R^2$) are included within the figures, both for EXT ensemble (gray) and ALL ensemble (red). Solid lines indicate that correlations are significant (t-test with p<0.05). **(c,d)** Same as in Fig. 8a,b, but for the Atlantic Meridional Overturning Circulation (AMOC) index. **(e,f)** Sea Surface Temperature (SST) difference between extratropical ocean areas of the NH (EN) and extratropical ocean areas of the SH (ES) versus Atlantic OHT at 26ºN for the periods 2220-2239 and 2080-2099 with respect to the reference period 2020-2039, obtained with each simulation of **(e)** SSP5-3.4OS and **(f)** SSP1-1.9, as well as with the ensemble average of all the models (ALL), the ensemble average of simulations extended up to 2300 (EXT), and the ensemble average of each model containing several simulations.

not support a linear relationship between both quantities (Fig. 8f). The relationship between Atlantic OHT at 26ºN and both the

AMOC index (Fig. 8c,d) and the MLD in NNA (Fig. 8a,b) is almost linear, with significant correlations of 0.98 and 0.99 for SSP5-3.4OS EXT (Fig. 8a,c) and 0.79 and 0.46 for SSP1-1.9 ALL (Fig. 8b,d). Changes during the overshoot are significantly stronger for the models with the largest climatological values of MLD, with a $R^2$ coefficient between the post-overshoot MLD change and its reference climatology of 0.98 for SSP5-3.4OS EXT (Fig. 9a) and 0.52 for SSP1-1.9 ALL (Fig. 9b).

The ENM-ENH temperature asymmetry cannot be clearly linked to changes in the Atlantic OHT (Fig. 10a,b). In this case,

the contribution of sea ice changes (Fig. 10c,d) seems more important. Overall, simulations with a larger decline in the NH sea ice area after the overshoot, like those from UKESM1-0-LL and CMCC-ESM2 and some members from IPSL-CM6A-LR, CanESM5 and MIROC6, also show a strong ENM-ENH asymmetry. By contrast, simulations with weaker changes in the sea ice, like those from CNRM-ESM2-1, ACCESS-CM2, MPI-ESM1-2-LR, and FGOALS-g3, do not show relevant temperature asymmetries between the mid and high latitudes of the NH. Despite this consistent behavior, the $R^2$ coefficients when con-

sidering ensemble averages are small, with only 0.22 for SSP5-3.4OS ALL (Fig. 10c) and 0.31 for SSP1-1.9 ALL (Fig. 10d),



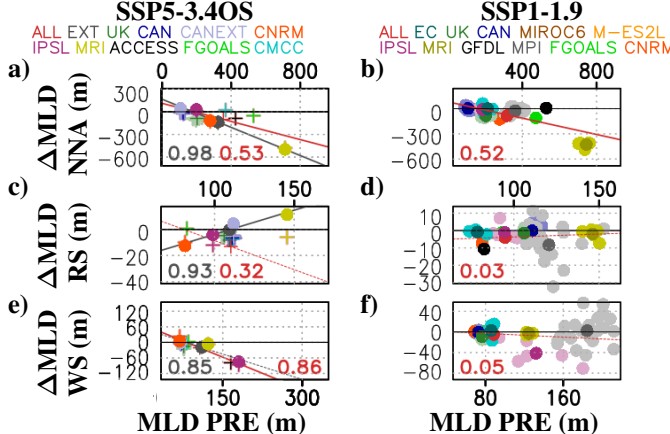

**Figure 9. (a,b)** March Mixed Layer Depth (MLD) in the northern North Atlantic (NNA) post-overshoot (2220-2239 and 2080-2099) to pre-overshoot (2020-2039) difference versus its climatology for the pre-industrial period (PRE; 1861-1880), obtained with each simulation of **(a)** SSP5-3.4OS and **(b)** SSP1-1.9, as well as with the ensemble average of all the models (ALL), the ensemble average of simulations extended up to 2300 (EXT), and the ensemble average of each model containing several simulations. Regression lines and coefficients of determination ($R^2$) are included within the figures, both for EXT ensemble (gray) and ALL ensemble (red). Solid lines indicate that correlations are significant (t-test with $p < 0.05$). **(c,d)** Same as in Fig. 9a,b, but for September MLD in Ross Sea (RS). **(e,f)** Same as in Fig. 9a,b, but for September MLD in Weddell Sea (WS).

evidencing important differences across the models. As shown in Fig. 10d, there are also important differences across the simulations from a given model, indicating that results can also be sensitive to internal variability. Interestingly, within some model ensembles, like those from CanESM5 and MIROC6, the simulations support a linear relationship between the changes in the sea ice area and the ENM-ENH asymmetries, indicating that in these models a relationship between sea-ice area and

SST is found associated with internal variability.

Unlike for the ENM-ENH asymmetry, the contribution of sea ice changes to the SSW-SSE asymmetry (Fig. 11i,j) seems marginal. Simulations with a larger increase of SH sea ice area, like those from CanESM5 and FGOALS-g3 in SSP1-1.9 (Fig. 11j), are not necessarily those with larger SSW-SSE asymmetries. In this case, the contribution of OHT changes seems more important (Fig. 11a,b). Those simulations with a post-overshoot state characterized by lower values of the northward Atlantic

OHT at 30ºS are also those showing the largest temperature asymmetries between eastern and western Southern Ocean. This is the case for MRI-ESM2-0 and CNRM-ESM2-1 SSP5-3.4OS simulations (Fig. 11a), and for the SSP1-1.9 ensemble of UKESM1-0-LL (Fig. 11b). A relevant contribution of the OHT at 50°S —which can be linked to the SMOC —to the SSW-SSE asymmetry is also found (Fig. 11g-h), as the models with a larger OHT increase (e.g. UKESM1-0-LL, IPSL-CM6A-LR, and CNRM-ESM2-1) tend to simulate the largest temperature asymmetries. For the case of UKESM1-0-LL, this situation is

also associated with a decrease of the MLD in the WS and RS regions (Fig. 11c-f), while for the IPSL-CM6A-LR the decrease is limited to the WS region (Fig. 11e,f), and for CNRM-ESM2-1 it is limited to the RS region (Fig. 11c,d), reflecting that the



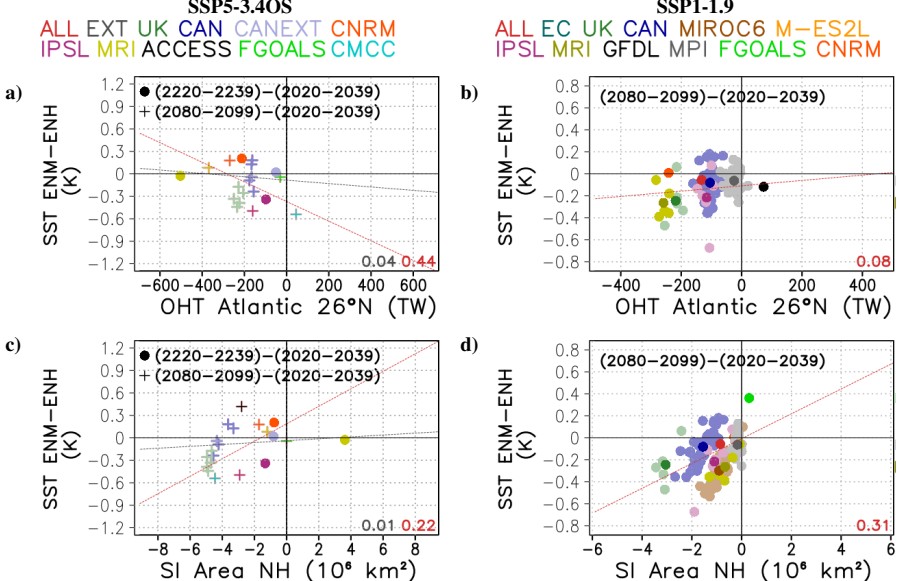

**Figure 10. (a,b)** Sea Surface Temperature (SST) difference between mid-latitude extratropical ocean areas of the NH (ENM) and high-latitude extratropical ocean areas of the NH (ENH) versus Atlantic Ocean Heat Transport (OHT) at 26ºN for the periods 2220-2239 and 2080-2099 with respect to the reference period 2020-2039, obtained with each simulation of **(a)** SSP5-3.4OS and **(b)** SSP1-1.9, as well as with the ensemble average of all the models (ALL), the ensemble average of simulations extended up to 2300 (EXT), and the ensemble average of each model containing several simulations. Regression lines and coefficients of determination ($R^2$) are included within the figures, both for EXT ensemble (gray) and ALL ensemble (red). Solid lines indicate that correlations are significant (t-test with p<0.05). **(c,d)** Same as in Fig. 10a,b, but for SST ENM-ENH differences versus Sea Ice (SI) area in the NH.

key areas of MLD in the Southern Ocean are highly model dependent. For the case of SSW-SSE asymmetry, the large spread among the models reduces the $R^2$ coefficient between OHT and temperature differences, to 0.43 for SSP5-3.4OS EXT (Fig. 11g) and to 0.17 for SSP1-1.9 ALL (Fig. 11h).

To better understand the spatial patterns associated with these changes, the post-overshoot state (2220-2239 for the SSP5-3.4OS EXT ensemble and 2080-2099 for the SSP5-3.4OS and SSP1-1.9 ALL ensembles) is compared with the pre-overshoot state (2020-2039) for all the models providing extended simulations of SSP5-3.4OS, namely, MRI-ESM2-0 (Fig. 12), CNRM-ESM2-1 (Fig. 13), IPSL-CM6A-LR (Fig. 14), and CanESM5 (Fig. 15). From these models, the simulations from MRI-ESM2-0 show very clear EN-ES asymmetries (Fig. 12a), while the simulations from CNRM-ESM2-1 show more prominent ENM-ENH

asymmetries (Fig. 13a), and the simulations from IPSL-CM6A-LR show larger SSW-SSE asymmetries (Fig. 14a).

The simulations from MRI-ESM2-0 (Fig. 12) are characterized by strong changes of the AMOC, reflected as strong changes in the Atlantic OHT between 30ºS and 30ºN (Fig. 12b), the Atlantic mass transport (Fig. 12c) and the MLD around Greenland (Fig. 12e). These strong changes, which can be linked to the particularly large climatological values of MLD in NNA simulated by this model, generate a strong temperature asymmetry between extratropical areas of the NH and SH (Fig. 12a), which can be



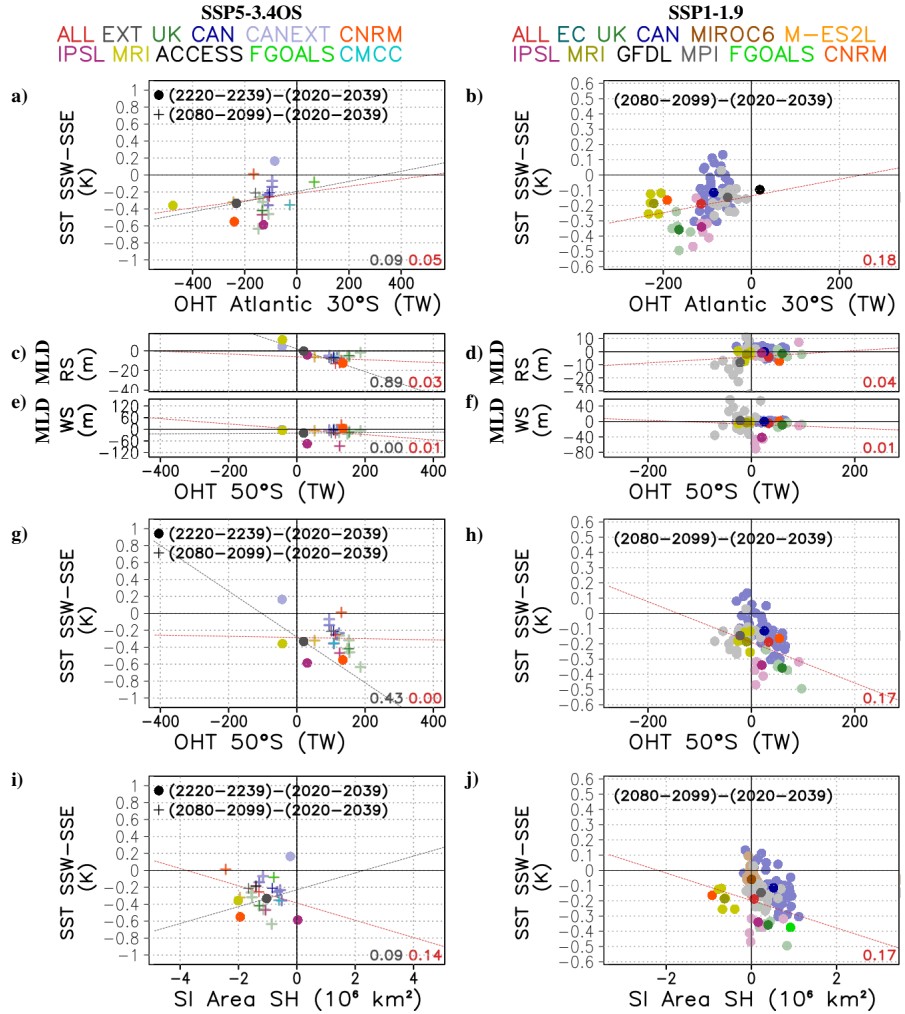

**Figure 11. (a,b)** Sea Surface Temperature (SST) difference between south-western Southern Ocean (SSW) and south-eastern Southern Ocean (SSE) versus Atlantic Ocean Heat Transport (OHT) at 30ºS for the periods 2220-2239 and 2080-2099 with respect to the reference period 2020-2039, obtained with each simulation of **(a)** SSP5-3.4OS and **(b)** SSP1-1.9, as well as with the ensemble average of all the models (ALL), the ensemble average of simulations extended up to 2300 (EXT), and the ensemble average of each model containing several simulations. Regression lines and coefficients of determination ($R^2$) are included for all the panels, both for EXT ensemble (gray) and ALL ensemble (red). Solid lines indicate that correlations are significant (t-test with p<0.05). **(c,d)** September Mixed Layer Depth (MLD) in Ross Sea (RS) versus global OHT at 50ºS for the periods 2220-2239 and 2080-2099 with respect to the reference period 2020-2039, obtained with each simulation of **(c)** SSP5-3.4OS and **(d)** SSP1-1.9, as well as with the ensemble average of all the models (ALL), the ensemble average of simulations extended up to 2300 (EXT), and the ensemble average of each model containing several simulations. **(e,f)** Same as in Fig. 11c,d, but for the September MLD in Weddell Sea (WS). **(g,h)** Same as in Fig. 11a,b, but for SST SSW-SSE differences versus global OHT at 50ºS. **(i,j)** Same as in Fig. 11a,b, but for SST SSW-SSE differences versus Sea Ice (SI) area in the SH.



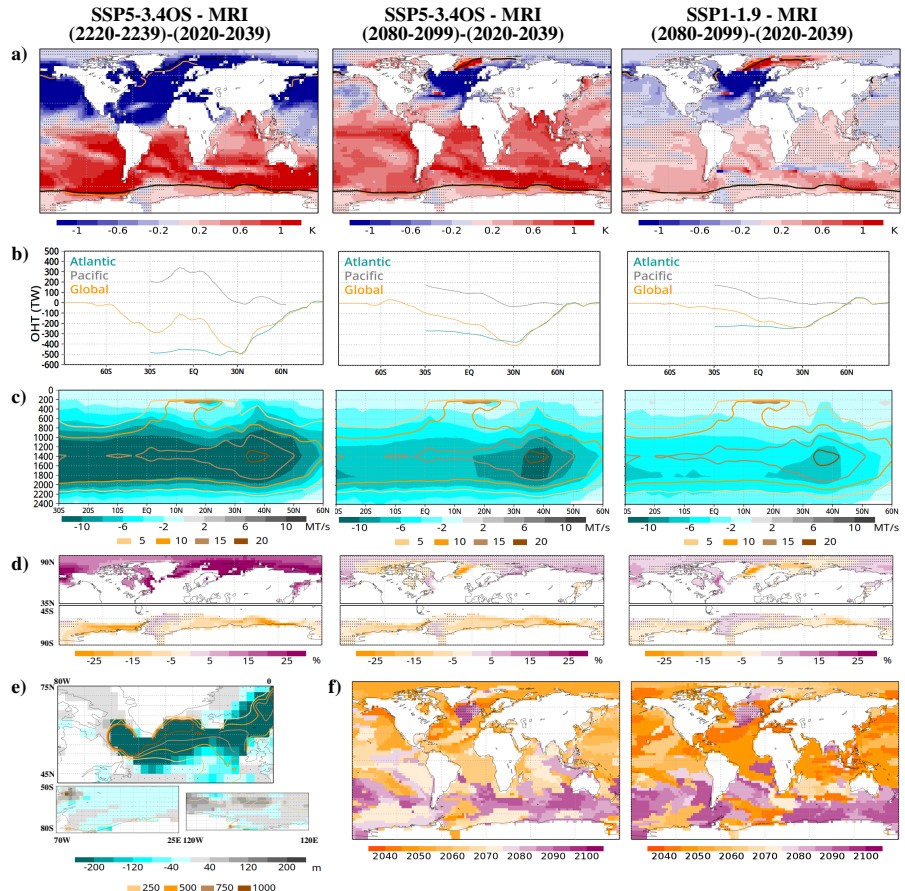

**Figure 12. (a-d)** Difference between the ensemble mean, temporal average values of **(a)** Sea Surface Temperature (SST), **(b)** Ocean Heat Transport (OHT; positive to the North), **(c)** Atlantic mass transport (positive to the South), and **(d)** sea ice concentration for **(left)** the periods 2220-2239 and 2020-2039 obtained with the MRI-ESM2-0 simulation of SSP5-3.4OS, **(center)** the periods 2080-2099 and 2020-2039 obtained with the MRI-ESM2-0 simulation of SSP5-3.4OS, and **(right)** the periods 2080-2099 and 2020-2039, obtained with the MRI-ESM2-0 ensemble of SSP1-1.9. Contours of 15% of sea ice concentration are included in the maps of SST, both for the periods 2220-2239 and 2080-2099 (yellow) and for the reference period 2020-2039 (black). Differences of OHT are shown per latitude, for the Atlantic basin, the Pacific basin, and the global zonal average. Differences of Atlantic mass transport are shown per latitude and level. Contours indicate climatological values of Atlantic mass transport obtained with the period 1861-1880. Stippling indicates locations where the differences are not significant (t-test with p<0.05). **(e)** Difference between the ensemble mean, temporal average values of March Mixed Layer Depth (MLD) in the northern North Atlantic (NNA) and September MLD in Weddell Sea (WS) and Ross Sea (RS) for the periods 2220-2239 and 2020-2039 obtained with the MRI-ESM2-0 simulation of SSP5-3.4OS. Contours indicate climatological values of MLD obtained with the period 1861-1880. Stippling indicates locations where the differences are not significant (t-test with p<0.05). **(f)** Year of maximum SST for **(left)** the MRI-ESM2-0 simulation of SSP5-3.4OS and **(right)** the MRI-ESM2-0 ensemble of SSP1-1.9. Stippling indicates locations where the overshoot is associated with a temperature minimum. In those cases, the year of the minimum is considered instead.



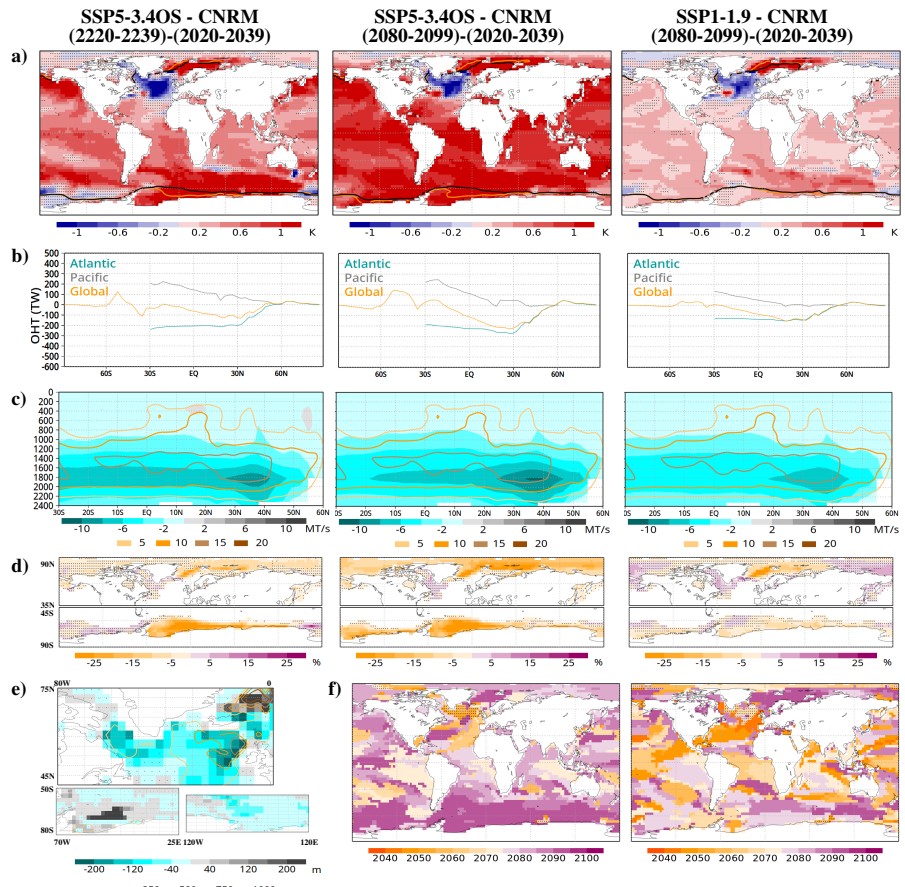

**Figure 13.** Same as Figure 12, but for the CNRM-ESM2-1 model.

at least partly explained by a different timing of the temperature maximum (Fig. 12f). In the NH, an asymmetry is also found between mid-latitude areas (with lower temperatures after the overshoot) and high-latitude areas (with higher temperatures instead) even if this ENM-ENH asymmetry is less prominent than the EN-ES asymmetry. The strong temperature asymmetry between NH and SH impacts also the sea ice distribution (Fig. 12d), which is generally increased in the NH and decreased in the SH. Only some areas of the NH around Greenland and Barents Sea show a relative decrease for the case of SSP1-1.9 and SSP5-3.4OS up to 2100.

The simulations from CNRM-ESM2-1 (Fig. 13) are characterized by strong sea ice changes (Fig. 13d). Both for the NH and SH, the sea ice lost during the $CO_2$ increasing phase is not fully recovered during the $CO_2$ decreasing phase (Fig. 13d), generating higher ocean surface temperatures in the polar areas after overshoot (Fig. 13a). This temperature asymmetry between mid-latitude and polar areas is also reflected in the timing of the temperature maximum (Fig. 13f), which is delayed in the polar areas of both hemispheres. Even if smaller in magnitude than those of MRI-ESM2-0, relevant changes are also found in the AMOC and its associated variables, with lower northward Atlantic OHT after overshoot (Fig. 13b), a decrease of the Atlantic





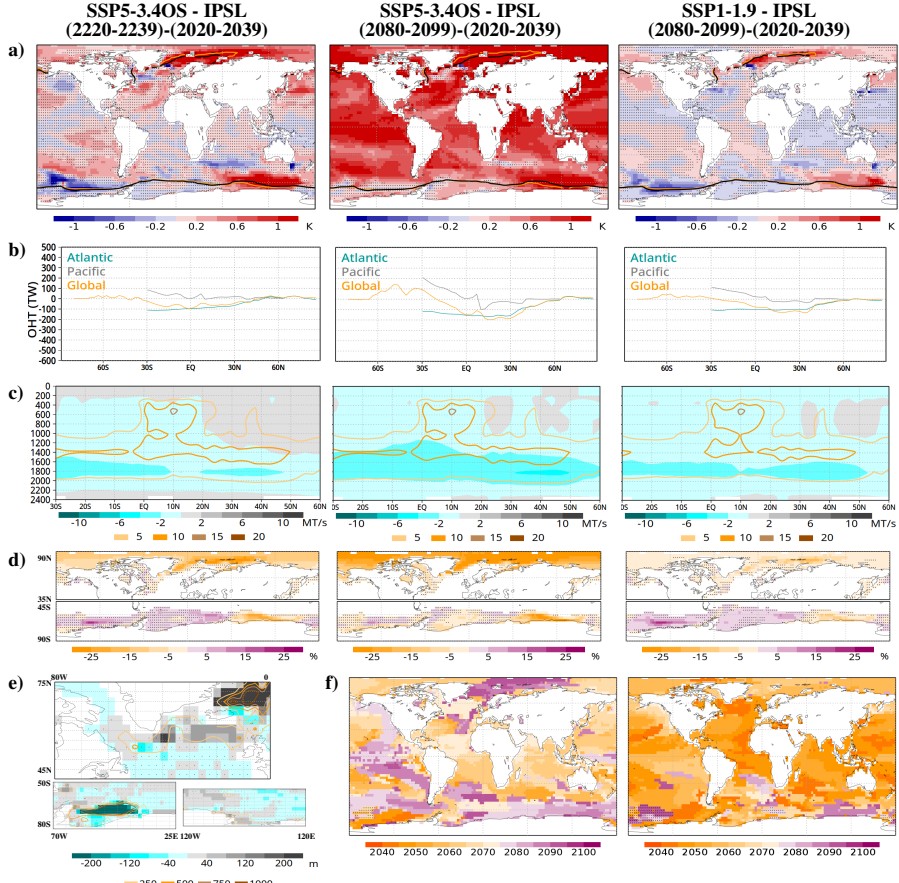

**Figure 14.** Same as Figure 12, but for the IPSL-CM6A-LR model.

mass transport (Fig. 13c) and a lower MLD around Greenland, extending up to the North Sea (Fig. 13e). The smaller changes with respect to MRI-ESM2-0 can be linked to smaller climatological values of MLD in NNA. Changes of MLD are limited to those areas with the largest climatological values, including a decrease of MLD in some areas of Labrador sea and the eastern

subpolar North Atlantic that could explain the AMOC weakening, and an increase north of Iceland, potentially associated with a lack of sea ice recovery after overshoot. In the Southern Ocean, the OHT at 50ºS is slightly increased in the post-overshoot situation (Fig. 13b), and a small increase of MLD is found in the WS area, where sea ice experiences a post-overshoot decline (Fig. 13c).

     The simulations from IPSL-CM6A-LR (Fig. 14) are characterized by important changes in the Southern Ocean, with an

increase of the OHT at 50ºS (Fig. 14b) and an important decrease of the MLD in the WS, the Southern Ocean region with the largest climatological values (Fig. 14e). These regional changes in the Southern Ocean may explain higher temperatures in SSE and lower temperatures in SSW after overshoot (Fig. 14a), associated with a delayed temperature maximum in SSE with respect to SSW (Fig. 14f). The SSW-SSE temperature asymmetry can be linked to an asymmetry in the sea ice concentration,



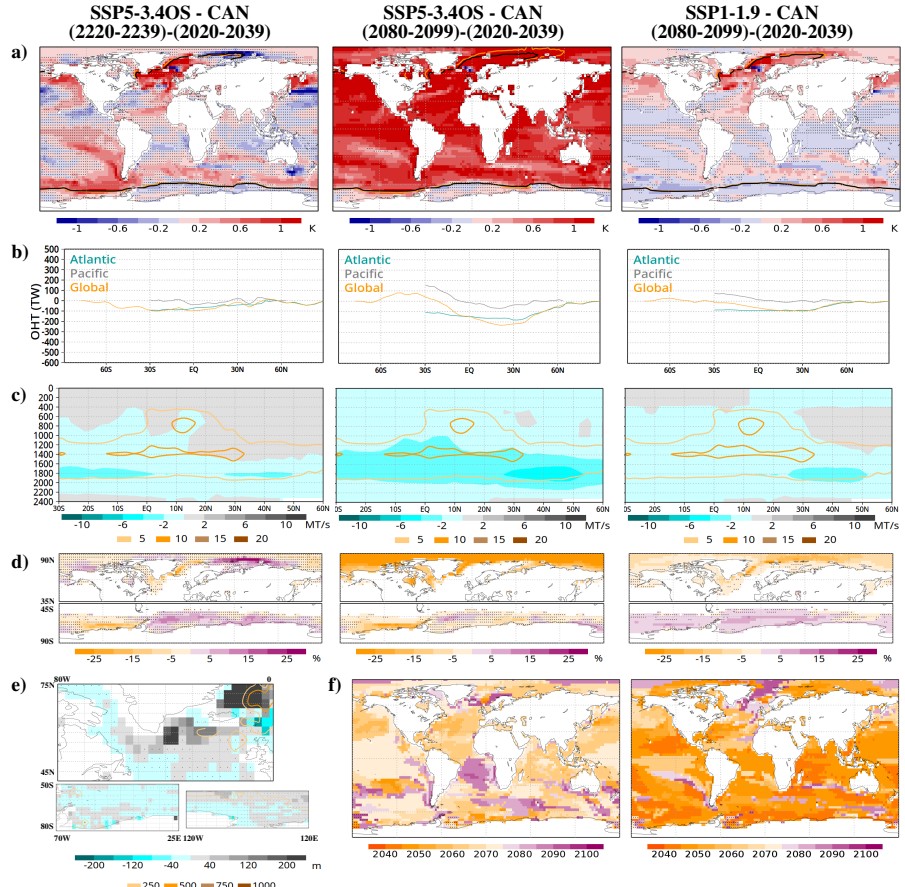

**Figure 15.** Same as Figure 12, but for the CanESM5 model.

which experiences a decrease for SSE and an increase for SSW after overshoot (Fig. 14d). In the NH, there is a widespread
decrease of sea ice concentrations (Fig. 14d), in line with the results from CNRM-ESM2-1 (Fig. 13d). The simulations from
IPSL-CM6A-LR present very low climatological values of MLD in NNA, that experience a moderate strengthening after the
overshot. These MLD changes cannot thus explain the weakening observed for the AMOC after the overshoot (Fig. 14c),
accompanied by a decrease of the northward Atlantic OHT (Fig. 14b), changes that are, however, smaller in magnitude than
those simulated by MRI-ESM2-0 and CNRM-ESM2-1.

The simulations from CanESM5 (Fig. 15) also show relevant changes in the Southern Ocean, with an increase of the OHT
at 50ºS (Fig. 15b) similar to that of IPSL-CM6A-LR (Fig. 14b) but without a clear strengthening of the MLD in the WS (Fig.
15e). In the NH, an ENM-ENH asymmetry is also found (Fig. 15b), potentially associated with a lack of recovery of NH
sea ice after overshoot, as for the case of CNRM-ESM2-1 (Fig. 13d). As for the case of IPSL-CM6A-LR, the simulations of
CanESM5 present low climatological values of MLD in NNA that are moderately strengthened after the overshoot, thus again
decoupled from the observed weakening of the AMOC (Fig. 15c) and the northward Atlantic OHT (Fig. 15b), both smaller




than for MRI-ESM2-0 and CNRM-ESM2-1. As for the case of IPSL-CM6A-LR, the retreat of sea ice coverage after overshoot increases the MLD in some areas of the eastern North Atlantic.

## 4    Discussion and Conclusions

The analysis of CMIP6 overshoot scenarios (SSP5-3.4OS and SSP1-1.9) confirms that certain regions do not fully return to
their pre-overshot state. This incomplete recovery is marked by persistent large-scale temperature asymmetries, associated with differences in the timing of the temperature maximum (Roldán-Gómez et al., 2025). Important temperature asymmetries are found between extratropical areas of the NH and SH, as well as between mid and high latitudes of the NH, and between eastern and western areas of the Southern Ocean. The relative role of these temperature asymmetries strongly depends on the model, with some models like MRI-ESM2-0 presenting strong asymmetries between NH and SH, some others like CNRM-ESM2-
1 asymmetries between mid and high latitudes, and some models like IPSL-CM6A-LR and CanESM5 showing important asymmetries in the Southern Ocean.

These temperature asymmetries can be associated with several mechanisms introducing hysteresis in the climate system during the overshoot, including changes in ocean dynamics and meridional overturning circulation (Palter et al., 2018), both in the Atlantic and in the Southern Ocean (Li et al., 2024), as well as the melting of sea ice (Li et al., 2020). The post-overshoot
situation is generally characterized by a weakened AMOC, with lower northward OHT in the Atlantic and a decrease of MLD around Greenland. An alteration of the SMOC is also present in some simulations, with a decrease of MLD and southward OHT in the Southern Ocean. In addition, the sea ice lost during the $CO_2$ increasing phase is not always fully recovered during the $CO_2$ decreasing phase.

In line with Sgubin et al. (2017), the analyses performed in this work show that the contribution of each mechanism strongly
depends on the model, with some models like MRI-ESM2-0 showing strong alterations of the AMOC, some models like CNRM-ESM2-1 highlighting persistent changes in the sea ice, and some others like IPSL-CM6A-LR and CanESM5 simulating a relevant impact of the SMOC. These discrepancies between models can be associated with different climatological values for the MLD in the Subpolar North Atlantic region and in the Southern Ocean depending on the model. Models showing the largest climatological MLD also allow larger MLD decreases, producing a stronger AMOC slowdown, and then a larger hysteresis
in the meridional overturning during the overshoot. Identifying such relationships between climatological characteristics and projected changes could enable potential emergent constraints (Hall et al., 2019), if a robust observational estimate of MLD could be obtained. The contribution of each mechanism may also depend on the individual simulation of a given model, suggesting a relevant role of internal variability. This is found for example in Fig. 10d, in which different ensemble members of CanESM5 and MIROC6 show a different relative contribution of NH sea ice changes to the ENM-ENH asymmetry.

Despite these differences across models and simulations, the analyses show a link between EN-ES, ENM-ENH, and SSW-SSE temperature asymmetries and ocean dynamics and sea ice changes. Simulations with the largest EN-ES asymmetries are also those with the largest reductions in the Atlantic OHT, MLD in NNA region and AMOC index, while simulations



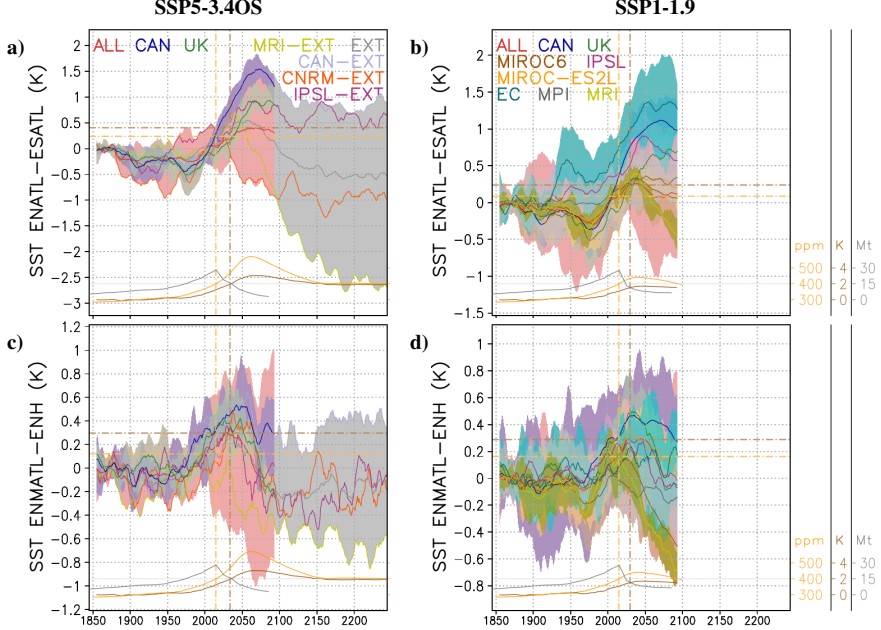

**Figure A1. (a,b)** Same as Fig. 2a,b, but for Atlantic extratropical areas of the NH (ENATL; 23° N - 90° N; Atlantic basin) and Atlantic extratropical areas of the SH (ESATL; 90° S - 23° S; Atlantic basin). **(c,d)** Same as Fig. 2a,b, but for Atlantic mid-latitude extratropical areas of the NH (ENMATL; 23° N - 60° N; Atlantic basin) and high-latitude extratropical areas of the NH (ENH; 60° N - 90° N).

showing important ENM-ENH asymmetries are those including a more limited recovery of NH sea ice after the overshoot, and simulations with the largest SSW-SSE asymmetries are also those with the largest changes in the OHT in the Southern Ocean.

The results of this study allow for a better understanding of the irreversibility of temperature in overshoot scenarios, by identifying the main large-scale temperature patterns present in the post-overshoot climate, as well as the mechanisms behind them. They also show substantial inter-model differences in the relative contributions of meridional overturning circulation and sea ice changes in case of overshoot associated in particular with the climatological MLD considered by each model for areas around Greenland and in the Southern Ocean. This paper sheds light on the processes associated with a variety of model

responses to the same forcing, in this case overshoot scenarios. Different processes dominate in different models, and the strength of changes in some specific features such as AMOC or MLD is associated with their model-specific mean state. Such in-depth analysis of changes and associated processes across models is the basis to better understand projected changes and can ultimately help to reduce the uncertainty of expected changes.

## Appendix A: Temperature asymmetries for Atlantic basin

Temperature asymmetries between NH and SH and between mid and high latitudes of the NH have been analyzed based on the EN, ES, ENM and ENH regions defined in Fig. 1. These asymmetries have been partly associated with changes in the AMOC,





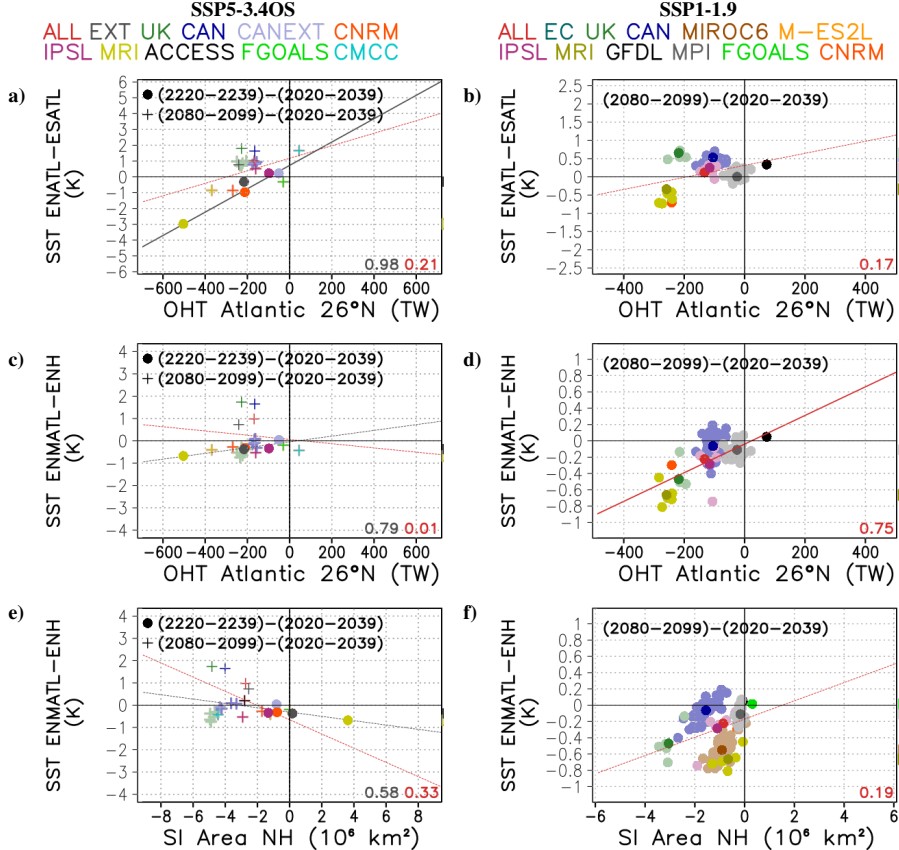

**Figure A2. (a,b)** Same as Fig. 8e,f, but for Atlantic extratropical areas of the NH (ENATL; 23° N - 90° N; Atlantic basin) and Atlantic extratropical areas of the SH (ESATL; 90° S - 23° S; Atlantic basin). **(c,d)** Same as Fig. 10a,b, but for Atlantic mid-latitude extratropical areas of the NH (ENMATL; 23° N - 60° N; Atlantic basin) and high-latitude extratropical areas of the NH (ENH; 60° N - 90° N). **(e,f)** Same as Fig. 10c,d, but for ENMATL and ENH.

mainly focused on the Atlantic basin. To verify that the results are not sensitive to the selection of the regions, and in particular to the use of regions covering all basins, the same analyses have been performed only for the Atlantic basin. In particular, Fig. A1 and A2 show the same results as for Fig. 2, 8 and 10 but considering the regions ENATL, ESATL and ENMATL. These

regions are equivalent to EN, ES and ENM but considering only the Atlantic basin. For ENH, as this region mainly covers the Arctic Ocean, the per-basin separation is not meaningful.

Figure A2c,d shows a larger impact of the Atlantic OHT at 26ºN in the ENMATL-ENH asymmetry than that obtained for the ENM-ENH asymmetry in Fig. 10a,b. The $R^2$ coefficient between OHT and temperature difference reaches 0.79 for SSP5-3.4OS EXT (Fig. A2c) and 0.75 for SSP1-1.9 ALL (A2d). Despite this minor difference, the results shown in Fig. A1 and A2

are generally in line with those from Fig. 2, 8 and 10, with ENATL-ESATL asymmetries behaving in a similar way as EN-ES asymmetries and ENMATL-ENH asymmetries behaving in a similar way as ENM-ENH asymmetries.





*Author contributions.* PJRG contributed with conceptualization of the study, data processing, discussion of results, and writing of the paper. PO and MGD contributed with discussion of results and writing of the paper.

*Competing interests.* The authors declare that they have no conflict of interest.

*Acknowledgements.* This research contributes to the Horizon 2020 project RESCUE (grant agreement No. 101056939) and the Spanish Ministry for Science and Innovation project PRECEDE (Grant No. EUR2022-134059). The authors are grateful to Margarida Samso-Cabre for downloading and formatting the CMIP6 input data used in the analyses.



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
