# Peer review of "Contribution of meridional overturning circulation and sea ice changes to large-scale temperature asymmetries in CMIP6 overshoot scenarios"

_EGUsphere, 2025_

## Author Comment (AC1)

**Responses to reviewer's comments for "Contribution of meridional overturning circulation and sea-ice changes to large-scale temperature asymmetries in CMIP6 overshoot scenarios"**

**Reviewer 1:**

We are grateful to the reviewer for the comments and suggestions, all of which have been helpful for improving the manuscript. We answer to each of the comments below, providing in gray the comments from the review and in black our responses.

Review of 'Contribution of meridional overturning circulation and sea -ice changes ot large-scale temperature asymmetries in CMIP6 overshoot scenarios'. By Roldán-Gómez
An analysis is conducted three different overshoot scenarios from 13 different coupled models. The focus is determining the spatial patterns of SST anomalies that emerge in the overshoot period. They find different spatial patterns emerge depending on whether there are large changes in the AMOC or sea-ice. The AMOC influence is greater when there is a larger reduction in NA mixed layer. This happens when the models have a larger climatological MLD to start with. These findings are in the main well-reasoned and the work should make a useful contribution to the literature. It is noted that the paper represents a considerable amount of analysis across multiple models to reach these conclusions. Indeed, aside of minor specific comments, my main general comment is that the authors need to make a greater effort to distil the research (and in particularly the number of figure panels) into that which is necessary to reach these conclusions.

**R1C1**

General Comments
A greater effort should be made to distil the research into that which is necessary to convince the reader of their conclusions. In particular, there is an overabundance of figures.
Although only 15 figures, this is made up of over 140 figure panels. I would encourage the authors to aggressively cull this number. There are a number of ways to evaluate if they are all necessary. Some of the plots contain linear regression, many of these could be probably best summarized by a table of the correlation values (and if they are significant or not). Furthermore, the authors may want to consider if they need to include results from all the scenarios. In particularly do the additional insights obtained from SSP1-1.9 warrant its inclusion.

Considering this comment, we have removed the following panels and the associated references in the text:
- Figure 2d, providing results consistent with those from Roldán-Gómez et al. (2025)
- Figure 5c-d, providing similar results to Fig. 5a-b
- Figure 6, since the link between OHT, AMOC and MLD is already illustrated in Fig. 8 and Fig. 11
- Figure 8c-d, providing relationship between AMOC and OHT
- Figure 9c-f, not discussed in the text
- Figure 11a-b, providing consistent results with those from Fig. 11g-h
- Figure 12f, 13f, 14f and 15f, not relevant for the discussion

Even if the results may be similar to those of SSP5-3.4OS, we still consider that the results for SSP1-1.9 are relevant, since this experiment includes more models and simulations than SSP5-3.4OS, and allows for a more complete assessment of model discrepancies. In addition, it also allows for assessing the role of internal variability, since it contains large ensembles of simulations, like those from CanESM5, MIROC6 and MPI-ESM1-2-LR.

For the figures with regressions, as discussed in the answer to R1C22, the multi-model regression is important, but it is not the only information provided by these figures. The behavior of each individual model is also interesting, and would be lost if the figures are replaced by tables with regression coefficients.

**R1C2**
Further Points
Abstract
Line 8: In the abstract. Candidate mechanisms are listed: 'Changes in the sea ice, changes in the ocean circulation and heat transport, and thermal inertia of the ocean have been identified as potential sources of hysteresis '. It would be helpful when listing these to enumerate these and when listing them elsewhere in the manuscript, to use consistent terminology.  (For example, later in the manuscript 'AMOC' is used rather than 'ocean circulation and heat transport ' ). This will help the reader relate the analysis in the manuscript with the abstract.  For example, the term 'Thermal inertia' is not mentioned in the manuscript beyond the introduction.

We have replaced "Changes in the sea ice, changes in the ocean circulation and heat transport, and thermal inertia of the ocean" by "Changes in the sea ice, the Atlantic Ocean Heat Transport (OHT) and the Atlantic Meridional Overturning Circulation (AMOC),", and have kept them elsewhere to use consistent terminology in the abstract and in the rest of the manuscript. We have removed the mention to thermal inertia, as it has not been addressed in our analyses.

**R1C3**
Line 30. A figure illustrating  the time series of CO2 concentration in two scenarios, would help the initiated.

Time series of $CO_2$ concentration for the two scenarios are included in Fig. 2a-b (yellow curves).

**R1C4**
Line 65. In light of the range of processes mentioned in the existing literature (line 40-44) why have the authors decided to focus on the "ocean dynamics and sea-ice changes"?

The work has been focused on the ocean dynamics and sea ice changes because of the large uncertainties existing for these two factors, as detailed in the introduction: "However, there are large uncertainties in these changes, with responses of the AMOC to forcing changes that strongly depend on the model considered (Sgubin et al., 2017)."

We have added a clarification on this: "This work analyzes overshoot scenarios from CMIP6

(SSP5-3.4OS and SSP1-1.9) to better understand why models present a different level of irreversibility in ocean dynamics and sea ice changes, and how this translates into the emergence of different large-scale temperature asymmetries in the post-overshoot climate."

**R1C5**

Methods
What is guiding the choice of simulations? Is it just everything available? The authors may consider if the inclusion of SP1-1.9 adds significantly to the work.

Yes, all the available simulations for all the CMIP6 experiments including overshoot (SSP5-3.4OS and SSP1-1.9) were considered in the analysis.

We have added a clarification on this: "The analyses are based on all the available simulations of CMIP6 overshoot experiments SSP5-3.4OS and SSP1-1.9, listed in Table 1"

As discussed in the answer to R1C1, we still consider that the results for SSP1-1.9 are relevant, since this experiment includes more models and simulations than SSP5-3.4OS, and allows for a more complete assessment of model discrepancies. In addition, it allows for assessing the role of internal variability, since, unlike SSP5-3.4OS, it contains large ensembles of simulations, like those from CanESM5, MIROC6 and MPI-ESM1-2-LR.

**R1C6**

Line 93. Could the authors state here how the post peak period is defined.

We have clarified the definition of post-overshoot period in that paragraph: "The post-overshoot state is defined as a 20-year period after stabilization for SSP5-3.4OS EXT (from 2220 and 2239) and as the last 20 years of simulations for the SSP1-1.9 and the SSP5-3.4OS ALL ensemble (from 2080 to 2099)"

**R1C7**

Line 110. As before, please clarify the reason for the choice of mechanisms investigated.

As stated in the answer to R1C4, these mechanisms have been selected because there are known discrepancies among the models.

We have added a clarification on this: "In a second step, some mechanisms and processes potentially contributing to these temperature asymmetries are explored. Considering the large model differences existing for these variables (Sgubin et al., 2017), the analyses are focused on the global, Atlantic and Pacific OHT, the Atlantic mass transport, the sea ice concentration, and the Mixed Layer Depth (MLD) provided by the models."

**R1C8**

Figures 2ab, 3, 5, 6, and 7. These figures are quite complicated. Could the authors provide a bit primer, so the readers can clearly identify the key points from them.

We have more specifically pointed out in the text to the relevant time series, instead of to the whole figures, as follows:

- "For example, for the extended simulation of MRI-ESM2-0 in SSP5-3.4OS (gold line in Fig. 2a) and for the ensemble average of MIROC6 (brown line in Fig. 2b), MIROC-ES2L (orange line in Fig. 2b), MPI-ESM1-2-LR (dark grey line in Fig. 2b), and MRI-ESM2-0 (gold line in Fig. 2b) in SSP1-1.9, the global average of SST after overshoot is lower than that of before."

- "For example, the simulations with MRI-ESM2-0 show a EN-ES asymmetry larger than 2ºC at the end of SSP5-3.4OS (year 2300; gold line in Fig. 3a) and close to 0.3ºC at the end of SSP1-1.9 (year 2100; gold line in Fig. 3b), while for other simulations like those with IPSL-CM6A-LR (purple line in Fig. 3a,b) this asymmetry is not clearly found."

- "this is only found for some simulations like those from MRI-ESM2-0 and IPSL-CM6A-LR (gold and purple lines in Fig. 3c,d), while some others like those from CNRM-ESM2-1 (dark orange line in Fig. 3c) or MIROC-ES2L (orange line in Fig. 3d) do not show a clear asymmetry. Less discrepancies exist for the SSW-SSE asymmetry (Fig. 3e,f), for which most model simulations show colder temperatures in the western sector after the overshoot, even if some particular simulations like those from EC-Earth3 show comparable temperatures in both regions (turquoise line in Fig. 3f)."

- "As for the case of temperatures, the temporal evolution of OHT for individual models (Fig. 5) highlights a strong inter-model dispersion. MRI-ESM2-0 shows a strong decrease of Atlantic OHT (gold line in Fig. 5a,b), while other models like CanESM5 and IPSL-CM6A-LR (blue and purple lines in Fig. 5a,b) show a more limited decrease. In the Southern Ocean, some models like CNRM-ESM2-1 (dark orange line in Fig. 5c) and IPSL-CM6A-LR (purple line in Fig. 5c,d) show a relevant strengthening of the northward OHT at the end of the run, while others like MRI-ESM2-0 (gold line in Fig. 5c,d) do not show relevant changes with respect to the pre-overshoot state."

- "As shown in Fig. 6a,c, the average of the SSP5-3.4OS EXT ensemble (gray line) shows a larger sea ice area in the NH and a smaller sea ice area in the SH after overshoot, but this mean response is dominated by the simulations from MRI-ESM2-0 (gold line in Fig. 6a,c), to a lesser extent, CNRM-ESM2-1 (dark orange line in Fig. 6a,c)"

As for the answer to comment R1C1, we have removed Figure 6. Figure 7 would then become Figure 6.

**R1C9**

Line 121. 'After the stabilization'. Stabilization of what. Could the authors refer to specific years.

In this context, "after the stabilization" means when the global temperature is already stabilized. This happens at a different time depending on the model and on the experiment.

To avoid any misunderstanding, we have replaced "After the stabilization" by "At the end of the simulations (2300 for the extended simulations of SSP5-3.4OS and 2100 for the others)".

**R1C10**

Line 122. 'Larger than expected'. It is not clear what is expected. Please state the years that are being referred to, what specifically is it larger than (and how much).

As stated in the text, it refers to "larger than the expected value considering the situation before the overshoot with the same global average of surface air temperature (horizontal brown line) or the situation before the overshoot with the same $CO_2$ concentration (horizontal yellow line)"

We have added the exact years in the text and to avoid the term "expected": "is larger than that of the situation before the overshoot with the same global average of surface air temperature (horizontal brown line; reached in 2034 for SSP5-3.4OS and in 2030 for SSP1-1.9) or the situation before the overshoot with the same $CO_2$ concentration (horizontal yellow line; reached in 2015 both for SSP5-3.4OS and SSP1-1.9)"

**R1C11**

Line 150. MRI shows a 2.0 C EN-ES asymmetry at year 2200 for SSP5-4.4OS (Figure 3a). However, for SSP1.-19, the maximum asymmetry is only about 0.6C at about 2040 (Figure 3b)". Unless I fail to understand the figure.

Yes, this is true. The comment was mostly refering to the relative value of MRI with respect to the other models. For both experiments, at the end of the run the simulations from MRI show the largest changes. But indeed, when comparing both experiments, the changes are much more relevant for SSP5-3.4OS, which extend much further in time.

We have reworded the sentence as: "The comparison of different models (Fig. 3) for indices that represent the aforementioned asymmetries reveals that these asymmetries are not equally present in all models. For example, the simulations with MRI-ESM2-0 show a EN-ES asymmetry larger than 2ºC at the end of SSP5-3.4OS (year 2300; gold line in Fig. 3a) and close to 0.3ºC at the end of SSP1-1.9 (year 2100; gold line in Fig. 3b), while for other simulations like those with IPSL-CM6A-LR (purple line in Fig. 3a,b) this asymmetry is not clearly found."

**R1C12**

Line 150 'Strong'. Please define 'strong' in turns of magnitude and state the time at which this can be seen in the relevant figure.

We have included the magnitude of the asymmetry and the time at which this can be seen (end of simulations): "The comparison of different models (Fig. 3) for indices that represent the aforementioned asymmetries reveals that these asymmetries are not equally present in all models. For example, the simulations with MRI-ESM2-0 show a EN-ES asymmetry larger than 2ºC at the end of SSP5-3.4OS (year 2300; gold line in Fig. 3a) and close to 0.3ºC at the end of SSP1-1.9 (year 2100; gold line in Fig. 3b), while for other simulations like those with IPSL-CM6A-LR (purple line

in Fig. 3a,b) this asymmetry is not clearly found."

**R1C13**

3.1 Line 150-156. What does the variety in the results reveal?

As discussed in section 4, the different temperature asymmetries shown by the models are indicative of a different representation of ocean dynamics and sea ice mechanisms in the models. Some models like MRI-ESM2-0 show strong alterations of the AMOC, some models like CNRM-ESM2-1 highlight persistent changes in the sea ice, and some others like IPSL-CM6A-LR and CanESM5 simulate a relevant impact of the SMOC. These discrepancies between models can be associated with different climatological values for the MLD in the Subpolar North Atlantic region and in the Southern Ocean depending on the model. Models showing the largest climatological MLD also allow larger MLD decreases, producing a stronger AMOC slowdown, and then a larger hysteresis in the meridional overturning during the overshoot.

It is preferred to keep this discussion in section 4, and keep section 3.1 only for the presentation of temperature results.

**R1C14**

Line 173. SO higher mean values of OHT are found in SSP5-3.4OS-ALL (middle panel) but not SSP5-3.4OS EXT (left) or SSP1-1.9 ALL (right)

Right, this is a mistake. We have corrected this as: "In the Southern Ocean, higher values of northward OHT are found on average between 60ºS and 30ºS for the SSP5-3.4OS ALL ensemble (Fig. 4a, center), but this is not the case for the SSP1-1.9 ALL ensemble (Fig. 4a, right) and the SSP5-3.4OS EXT ensemble (Fig. 4a, left)". Thank you for noticing this.

**R1C15**

Line 175. 'Some models like MRI-ES2.0'. The wording of this sentence is a little strange as figure Fig. 5 abcd show that MRI-ES2 is an outlier, i.e. there are no other models like it.

The wording has been changed to: "MRI-ESM2-0 shows a strong decrease of Atlantic OHT (gold line in Fig. 5a,b), while other models like CanESM5 and IPSL-CM6A-LR (blue and purple lines in Fig. 5a,b) show a more limited decrease."

**R1C16**

Line 177. 'moderate strengthening of OHT in SO in Fig 5ef'. Could the authors specify the post-overshoot years in which OHT is moderately strengthened.

It refers to the end of the simulations with respect to the pre-overshoot period with the same CO2 concentration.

To avoid any ambiguity, we have reworded the sentence as: "In the Southern Ocean, some models like CNRM-ESM2-1 (dark orange line in Fig. 5c) and IPSL-CM6A-LR (purple line in Fig. 5c,d)

show a relevant strengthening of the northward OHT at the end of the run, while others like MRI-ESM2-0 (gold line in Fig. 5c,d) do not show relevant changes with respect to the pre-overshoot state.

**R1C17**

Line 180. MLD Fig 5e. IPSL is strong outlier in terms of WS MLD. For SSP5-340S, it looks like the change in the other models is not so significant.

Indeed, the IPSL shows a behavior for the WS MLD very different from the other models. But this result is still relevant, since this shows that a model with strong changes in the WS MLD may also show strong changes in the OHT at 50ºS during the overshoot.

**R1C18**

Fig. 8. It is difficult to see difference between dotted and solid line, please thicken the solid line or strengthen the difference in some other way. Probably no need for AMOC vs OHT, the relationship is well established.

We have thickened the solid lines in Fig. 8, 9, 10 and 11 (new 7, 8, 9 and 10). To improve the readability of the figures, we have also replaced dotted lines by long-dashed lines.

Panels c-d have been removed from Fig. 8 (new 7), since the relationship between AMOC and OHT is well established.

The new figures would be then as follows:

New Fig. 7:

[Figure]

New Fig. 8:

[Figure]

New Fig. 9:

New Fig. 10:

[Figure]

**R1C19**

Line 199. Suggest 'Figure 8e and f) shows…'. The description of Figure 8 is not quite correct. EN-ES is only related to OHT. AMOC and MLD are related to OHT not to EN-ES SST. It might be more useful to have EN-ES vs OHT and EN-ES vs MLD.

We decided to use the OHT as pivot for the comparisons, since the OHT can be directly linked on one side to the EN-ES (changes in the heat transport physically relate to changes in temperature) and on the other side to the MLD (changes in the MLD may impact the AMOC, and then the OHT). Even if the links between EN-ES and OHT and between OHT and MLD indirectly imply a link between EN-ES and MLD, directly presenting EN-ES vs MLD may mask the physical process explaining the relationship between these two variables.

We have reworded the sentence as "Figure 7c,d shows how the EN-ES temperature asymmetry — as characterized by the difference between the post-overshoot (2220-2239 and 2080-2099) and pre-overshoot (2020-2039) states — in individual simulations and their ensemble mean relate to changes in Atlantic OHT at 26ºN."

The link between OHT and AMOC and MLD in NNA is commented later on in the paragraph: "The relationship between Atlantic OHT at 26ºN and the MLD in NNA (Fig. 7a,b) is almost linear, with significant correlations of 0.99 for SSP5-3.4OS EXT (Fig. 7a) and 0.46 for SSP1-1.9 ALL (Fig. 7b)"

**R1C20**
Fig 8f) What is the physical explanation for the lack of correlation between EN-ES SST and OHT in SSP1.1.9?

Looking at Fig. 3b, the EN-ES SST at the end of the SSP1-1.9 simulations does not seem fully stabilized, which suggests that they are still in a transient part of the post-overshoot state. Very likely this is the reason why the correlations between EN-ES SST and OHT are not emerging. If we focus for example on the simulations of MRI (with strong OHT and strong EN-ES SST in the SSP5-3.4OS experiment) we see that for SSP1-1.9 the EN-ES SST is still decreasing at the end of the run.

**R1C21**
There is no reference in text to Figure 9 c through f). I suggest these panels are removed.

As suggested, we have removed the panels c to f, and keep only the panels a and b.

**R1C22**
Line 215. 'The contribution of sea ice change seems more important'. Don't the dashed lines in Figure 10 indicate the contribution of sea-ice change is insignificant? I suggest the references to the influence of sea-ice are removed.

The dashed lines in the figure indicate that there is no significant correlation between sea ice area changes and ENM-ENH SST when considering all the models. However, when looking at the ensembles of individual models (e.g. SSP1-1.9 simulations of CanESM or MIROC6), those simulations with the largest changes in sea ice are also those showing the largest temperature asymmetry.

This has been discussed in section 4: "The contribution of each mechanism may also depend on the individual simulation of a given model, suggesting a relevant role of internal variability. This is found for example in Fig. 9d, in which different ensemble members of CanESM5 and MIROC6 show a different relative contribution of NH sea ice changes to the ENM-ENH asymmetry"

To avoid misunderstanding, we have reworded the sentence as: "The ENM-ENH temperature asymmetry cannot be clearly linked to changes in the Atlantic OHT (Fig. 9a,b). A certain contribution of sea ice changes (Fig. 9c,d) is found for some of the models. Indeed, in models like UKESM1-0-LL, CMCC-ESM2, IPSL-CM6A-LR, CanESM5 and MIROC6 the individual simulations with a larger decline in the NH sea ice area after the overshoot, show the strongest ENM-ENH asymmetry for those models."

**R1C23**

Line 226: 'Unlike for the ENM-ENH asymmetry, the contribution of sea ice changes to the SSW-SSE asymmetry (Fig. 11i,j) seems marginal'. Or like the ENM-ENH asymmetry it seems insignificant. If the authors do not wish to attach any importance to the significance test, they should probably explain why. The authors might consider if Figs, 10 and 11 are necessary.

As discussed in the answer to R1C22, the significance test shows that the relationship between sea ice changes and ENM-ENH asymmetry is not consistently found for all the models, but this does not prevent the changes in the sea ice to be relevant for certain models. This is for example the case for IPSL-CM6A-LR, CanESM5 and MIROC6.

This is discussed in the text: " Despite this consistent behavior, the R2 coefficients when considering ensemble averages are small, with only 0.22 for SSP5-3.4OS ALL (Fig. 10c) and 0.31 for SSP1-1.9 ALL (Fig. 10d), evidencing important differences across the models."

To avoid misunderstanding, we have removed "Unlike for the ENM-ENH asymmetry" from that sentence.

**Reviewer 2:**

We are grateful to the reviewer for the comments and suggestions, all of which have been helpful for improving the manuscript. We answer to each of the comments below, providing in gray the comments from the review and in black our responses.

Review of the manuscript: "Contribution of meridional overturning circulation and sea ice changes to large-scale temperature asymmetries in CMIP6 overshoot scenarios"
The manuscript, "Contribution of meridional overturning circulation and sea ice changes to large-scale temperature asymmetries in CMIP6 overshoot scenarios" by Pedro José Roldán-Gómez et al., presents an insightful investigation into the interplay between the Atlantic Meridional Overturning Circulation (AMOC), sea ice dynamics, and temperature patterns under overshoot climate scenarios. By analyzing a variety of CMIP6 models and focusing on different ensemble and temporal configurations, the study provides a nuanced understanding of the mechanisms shaping global and regional temperature asymmetries. I find the study extremely relevant for advancing our understanding of climate system responses to transient forcing. The use of multiple models and ensemble approaches enhances the robustness of the findings. The study is very valuable to the climate modeling and policy-making communities. However, I recommend a major revision, primarily due to issues with length, and figure overload that currently hinder the readability and accessibility of the results. I believe it requires significant revisions to enhance clarity, organization, and conciseness. Below, I outline both general and specific suggestions for improving the manuscript. I believe that, with a more concise and structured presentation, the manuscript could make a strong contribution to the literature on overshoot scenarios and climate feedback mechanisms.

**R2C1**

General Comments:

The "Results" section could benefit from clarification and reorganization. It is hard to read, as the authors compare multiple dimensions simultaneously: scenarios (SSP5-3.4OS and SSP1-1.9), pre- and post-overshoot periods, global averages and regional differences, EXT and ALL ensembles. It contains 13 figures, made of several plots and maps, which are overloading the manuscript. I suggest selecting the most relevant figures, increase their quality and illustrate the inter model results description in a more efficient and concise way.

We have reorganized the "Results" section as follows:

- Remove those figures that are not needed for the discussion, as indicated in the answers to R1C1, R2C16, R2C18 and R2C22.
- Better describe the pre- and post-overshoot periods, as indicated in the answers to R1C6 and R2C13.
- Move sea ice results to a dedicated subsection, as requested in comment R2C21.
- Move spatial patterns of individual models to Appendix B, as requested in comment R2C16.
- Re-organise the sections as: "Temperature changes", "Meridional overturning circulation changes", "Sea ice changes" and "Contributions to temperature changes across different models", as explained in the response to comment R2C15.

**R2C2**
Specific Comments:
Abstract:
Line 7: add a comma after "through it"

To address comment R2C4, we have done a reorganisation of the abstract, in which this comment is no longer applicable.

**R2C3**
Line 8: Remove one instance of "changes in"

The sentence has been reworded to address the comment R1C2 as: "Changes in the sea ice, the Atlantic Ocean Heat Transport (OHT) and the Atlantic Meridional Overturning Circulation (AMOC),"

**R2C4**
Line 10: The sentence beginning with "This work analyzes…" should be the first sentence of the abstract, describing the method before the results. Also, specify how many models are considered.

As requested, we have moved the sentence to the beginning of the abstract, with some slight rephrasing that now includes the number of models: "This work analyzes overshoot simulations from 13 models, including the SSP5-3.4OS and SSP1-1.9 experiments from the Coupled Model Intercomparison Project Phase 6 (CMIP6), to assess how regional temperatures after overshoot

differ from those of before, and how these differences may be impacted by changes in the sea ice and ocean circulation, including the Atlantic Ocean Heat Transport (OHT) and the Atlantic Meridional Overturning Circulation (AMOC). The overshoot scenarios, characterized by…"

The final part of the abstract has been adapted accordingly: "Changes in the sea ice and ocean circulation have been identified as potential sources of hysteresis, highlighting the impact of oceanic changes in the behavior of atmospheric variables in case of overshoot. The analyses from this work show that the relative contribution of each mechanism strongly depends on the model. Inter-model differences in the contributions of the meridional overturning can be associated with different climatologies of Mixed Layer Depth (MLD) in the northern North Atlantic (NNA) and in certain areas of the Southern Ocean. Despite these differences across models, both ocean circulation and sea ice changes contribute to shaping the regional temperatures after overshoot."

**R2C5**
Line 13-18: Mechanisms are described for only three models; this is too detailed for the abstract. The abstract also needs to be shortened — please remove these lines.

As suggested, we have removed the following sentence: "Certain models like MRI-ESM2-0 are mainly impacted by changes in the Atlantic Meridional Overturning Circulation (AMOC), others like CNRM-ESM2-1 show a relevant impact of sea ice changes in high-latitude areas, and others like IPSL-CM6A-LR show also a relevant impact of changes in the Southern Meridional Overturning Circulation (SMOC)."

**R2C6**
Introduction:
Line 1 and 3: The verb "increase" is used twice — consider rephrasing to avoid repetition.

We have rephrased as: "The growing probability of exceeding the temperature targets of the Paris Agreement of 2015 (Raftery et al., 2017), as a result of delays in implementing effective mitigation measures (IPCC, 2022), increases the likelihood of overshoot scenarios"

**R2C7**
Line 27: "is recovered" please rephrase to clarify

We have replaced "is recovered afterwards with net-negative emissions" by "is reduced to the target afterwards with net-negative emissions"

**R2C8**
Line 33: Please describe what "OS" in SSP5-3OS stands for. Also introduce "SSP"

We have included the meaning of OS and SSP, as: "Considering the increasing interest in these overshoots, the Coupled Model Intercomparison Project Phase 6 (CMIP6; Eyring et al., 2016) included in the ScenarioMIP (O'Neill et al., 2016) two scenarios with Shared Socio-economic Pathways (SSP) that reach a peak before starting a forcing decline: SSP5-3.4OS and SSP1-1.9. Both

of these represent Overshoot Scenarios (OS), but at different forcing levels."

**R2C9**

Line 61: Add this more recent paper, where transient forcing is used:
https://doi.org/10.1029/2025GL114611

We have added the reference as follows: "In global warming conditions, a decrease of OHT is generally found (Mecking and Drijfhout, 2023), both in the NH and in the SH, but models disagree on the presence of tipping points, with only some models showing a local convection collapse in the subpolar North Atlantic, and only a few of them presenting a full AMOC disruption (Sgubin et al., 2017). Changes in the AMOC also depend on the emission scenario and the associated loss of sea ice (van Westen and Baatsen, 2025). A collapse of the AMOC may have important impacts at a global scale (Orihuela-Pinto et al., 2022), stressing the need to understand in which conditions it may occur (Jackson et al., 2023). Baker et al. (2025) show that the collapse of the AMOC is unlikely even under extreme forcing conditions, and may only occur in case a strong Pacific Meridional Overturning Circulation (PMOC) emerges."

**R2C10**

Methods:
Line 74 and later: please change "have been" by "is" or "are", ex: "The analysis have been based" → "The analysis is based", more simple, direct

As suggested, we have done the following changes:
- "The analyses have been based" was replaced by "The analyses are based"
- "combined ensembles have been also considered" was replaced by "combined ensembles are also considered"
- "The ensemble averages of the ALL and EXT ensembles have been computed" was replaced by "The ensemble averages of the ALL and EXT ensembles are computed"
- "a longitude-latitude grid has been considered" was replaced by "a longitude-latitude grid is considered"
- "all the simulations have been remapped" was replaced by "all the simulations are remapped"
- "In a first step, analyses have been focused" was replaced by "In a first step, analyses are focused"
- "analyses have been based on temporal evolutions" was replaced by "analyses are based on temporal evolutions"
- "This post-overshoot state has been compared" was replaced by "This post-overshoot state is compared"
- "potential differences in aerosol emissions between SSP5-3.4OS and SSP1-1.9 have been assessed" was replaced by "potential differences in aerosol emissions between SSP5-3.4OS and SSP1-1.9 are assessed"
- "the regions included in Fig. 1 have been considered" was replaced by "the regions included in Fig. 1 are considered"
- "of the NH has been characterized" was replaced by "of the NH is characterized"
- "of the Southern Ocean has been characterized" was replaced by "of the Southern Ocean is characterized"

- "potentially contributing to these temperature asymmetries have been explored" was replaced by "potentially contributing to these temperature asymmetries are explored"
- "the analyses have been focused on the global" was replaced by "the analyses are focused on the global"
- "A set of indices have been defined" was replaced by "A set of indices are defined"
- "the temporal evolution of these indices have been evaluated" was replaced by "the temporal evolution of these indices are evaluated"
- "In particular, the asymmetry between NH and SH has been characterized" was replaced by "In particular, the asymmetry between NH and SH is characterized"

The same approach has been considered for the other sections:
- "different mechanisms potentially contributing to regional hysteresis in case of overshoot have been analyzed" was replaced by "different mechanisms potentially contributing to regional hysteresis in case of overshoot are analyzed"
- "Temperature asymmetries between NH and SH and between mid and high latitudes of the NH have been analyzed" was replaced by "Temperature asymmetries between NH and SH and between mid and high latitudes of the NH are analyzed"
- "These asymmetries have been partly associated" was replaced by "These asymmetries are partly associated"
- "the same analyses have been performed only for the Atlantic basin" was replaced by "the same analyses are performed only for the Atlantic basin"

**R2C11**
Line 84: Ocean outputs have also been regridded to 2.8 degrees? Why, since the coarsest model has a resolution of 1.7 degrees? Please justify.

The ocean outputs have been regridded to 2.8 degrees to have the same grid as for the atmospheric variables.

**R2C12**
Table 1: The first line of the caption is too long and redundant. When you mention "available simulations," do you mean that all available simulations on ESGF nodes were used? Please specify.

Yes, all the simulations available on ESGF nodes were considered.

We have simplified the caption to: "Table 1. Models and simulations considered in this work, resolution of atmospheric (ResA) and ocean component (ResO) of each model and associated references. For the SSP5-3.4OS, the simulations covering the extended period (up to 2300) are included in the column (EXT)."

**R2C13**
Line 93-99: The description of the periods is unclear; please consider adding a figure for illustration.

To also address comment R1C6, we have improved the description of the post-overshoot period as follows: "The post-overshoot state is defined as a 20-year period after stabilization for SSP5-3.4OS EXT (from 2220 and 2239) and as the last 20 years of simulations for the SSP1-1.9 and the SSP5-3.4OS ALL ensemble (from 2080 to 2099)"

The justification for the pre-overshoot period is already included in the following paragraph: "The post-overshoot state has been compared with the pre-overshoot state with the same global average of surface air temperature, which is reached in 2034 for SSP5-3.4OS and in 2030 for SSP1-1.9, and the pre-overshoot state with the same CO2 concentration (Meinshausen et al., 2020), reached in 2015 both for SSP5-3.4OS and SSP1-1.9. Considering these dates and in order to use a reference period large enough to focus on the long-term variability, our pre-overshoot reference for all analyses (except those stated otherwise) is the period from 2020 to 2039."

We prefer not to add a new figure, to avoid increasing the figure overload mentioned in comments R1C1 and R2C1.

**R2C14**

Figure 1: This figure is key; please make it twice as large. Some suggestions: add land in color, and draw the RS, WS, and NNA rectangles completely (over land as well).

Following the suggestion, land has been colored and the RS, WS and NNA rectangles have been completed.

[Figure]

We have used this new version as Figure 1 and increased the size to cover the whole text width.

**R2C15**

Results:
- The description of the results needs reorganization (see general comments). Inter-model

comparison results are discussed in the first subsection, yet another subsection is labeled "Differences across models".

Also, following the suggestions in comment R2C21, the sea ice results have been moved to the new section 3.3.

The section "Differences across models" mostly discusses the correlation between temperature asymmetries and circulation and sea ice changes, and how this correlation strongly depends on the model. In the previous sections, model differences are already discussed, but focusing on individual variables (temperature in section 3.1 and circulation and sea ice in section 3.2).

To avoid any misunderstanding, we have renamed the section 3.3 (now 3.4) as "Contributions to temperature changes across different models"

We have reformulated the first paragraph of this section as: "Considering the large discrepancies across the models in the simulation of temperature changes (Fig. 3), the OHT responses (Fig. 5), and the concomitant sea ice changes (Fig. 6), the contribution of each mechanism to large-scale temperature asymmetries must be analyzed on a per-model basis."

Another change has been implemented in response to comment R2C16. We have moved the discussion of spatial patterns for individual models to Appendix B.

**R2C16**

- The number, size, and selection of figures must be improved. Some suggestions are provided for Figure 2, which are expected to be applied to other figures (up to Figure 15) where possible. Consider moving Figures 12 to 15 - which focus on one model - to the appendix.

As suggested, we have moved Figure 12 to 15 to Appendix B, and rename them as Figure B1 to B4. We also removed the panel e (MLD) from these figures and included it in a new Figure B5.

We have also removed the panel f (year of the maximum) from these figures.

**R2C17**

- Figure 2: The size of the plots and maps is much too small; sea ice concentration is impossible to see, while the caption is unnecessarily long. Increase plot size, change the plot legend to use complete model names (e.g., "CAN" to "CanESM5"), and remove redundant explanations from the caption. Similarly, comparisons between periods need not be described in such detail in the caption.

We have increased the size of the figures to cover the whole text width and thus improve their visibility.

As suggested, we now included the complete model names within the figures and removed them from the caption.

The new figures would be then as follows:

New Fig. 2:

[Figure]

New Fig. 3:

[Figure]

New Fig. 5:

[Figure]

SSP5-3.4OS — ALL, UKESM1−0−LL, CanESM5, MRI−ESM2−0, CanESM5−EXT, CNRM−ESM2−1, IPSL−CM6A−LR, EXT

SSP1-1.9 — ALL, EC−Earth3, UKESM1−0−LL, CanESM5, MIROC6, MIROC−ES2L, IPSL−CM6A−LR, MRI−ESM2−0, MPI−ESM1−2−LR

a) OHT Atlantic 26°N (TW)
b) OHT Atlantic 26°N (TW)
c) OHT 50°S (TW)
d) OHT 50°S (TW)

New Fig. 6:

[Figure]

New Fig. 7:

[Figure]

New Fig. 8:

[Figure]

New Fig. 9:

New Fig. 10:

[Figure]

Fig. A1 and A2 are modified accordingly.

**R2C18**

Is the year of maximum SST really relevant? As you are comparing periods, consider moving this to the appendix.

We have removed the figures with the year of maximum SST, as well as the references to these figures in the text.

**R2C19**

The purpose of the vertical brown line is unclear. Are the vertical and horizontal lines relevant for discussion?

The horizontal lines indicate the level of the variable in the year with the same global air temperature (brown) and $CO_2$ concentration (yellow) as at the end of the simulation. This allows to compare the post-overshoot value (value at the end of the simulation) with the pre-overshoot value (horizontal line). Considering that we are analysing the irreversibility of variables, this difference

between post-overshoot and pre-overshoot values is more relevant than the absolute value of the variable. We consider then that the horizontal lines are relevant for the discussion.

Regarding the vertical lines, they help to understand how the horizontal lines are derived. They represent the year with the same global air temperature (brown) and $CO_2$ concentration (yellow) as at the end of the simulation. The pre-overshoot value (and then the horizontal lines) is defined with the point in which vertical lines cut the time series (red for the ALL ensemble and gray for the EXT ensemble).

**R2C20**
Please specify how the spread is evaluated.

The spread is directly the minimum/maximum value within the ensemble. No statistical metrics are computed.

This is now clarified in the figure caption: "The minimum-to-maximum spread comprised by the individual simulations within each ensemble is included with a shading"

**R2C21**
- Sea ice results should be moved to a specific subsection, rather than being merged with AMOC.

These results are now included in the new section "3.3. Sea ice changes"

The introductory paragraphs of sections 3.2 and 3.3 has been adapted as follows:

- "The results from Fig. 2 and Fig. 3 show persistent changes in the regional temperatures after the overshoot, but these results strongly depend on the model. To better understand the reasons behind this behavior, different mechanisms potentially contributing to regional hysteresis in case of overshoot are analyzed. Among these mechanisms, the alteration of the meridional overturning circulation plays a major role (Li et al.; 2024)."

- "Hysteresis in the sea ice coverage has been also identified as a mechanism potentially explaining irreversibility in temperature (Li et al.; 2020). As shown in Fig. 4c, the post-overshoot state for the SSP5-3.4OS EXT ensemble average is characterized by higher sea ice concentrations in the NH and lower sea ice concentrations in the SH (Fig. 4c, left), a feature that can be linked to the reduced global northward heat transport."

**R2C22**
- Figures 8 to 11 are very hard to read and contain 24 different plots. Please compress the information.

The following panels have been removed:

- Panels c-d in Fig. 8 (Fig. 7 in the new manuscript), as they provide consistent results with those

from panels 8a-b.
- Panels c-f in Fig. 9 (Fig. 8 in the new manuscript), as they are not discussed in the text.
- Panels a-b in Fig. 11 (Fig. 10 in the new manuscript), as they provide consistent results with those from panels 11g-h.

Figure 10 (Fig. 9 in the new manuscript) already contained fewer panels and could not be condensed further.

Also, following comment R2C17, the figures have been simplified, by putting the legend outside the figure.

**R2C23**
Discussion and Conclusions:
Discussion
The authors' interpretation of the AMOC–sea ice linkage would be strengthened by situating the results within key CMIP6 studies (Meccia et al, 2022) and discussing whether the model spread is robust across spatial domains.

We have added the following discussion on the AMOC-sea ice linkage: "These temperature asymmetries can be associated with several mechanisms introducing hysteresis in the climate system during the overshoot, including changes in ocean dynamics and meridional overturning circulation (Palter et al., 2018), both in the Atlantic and in the Southern Ocean (Li et al., 2024), as well as the melting of sea ice (Li et al., 2020). These processes are known to be inter-connected as revealed in the analysis of pre-industrial and global warming scenarios, with changes in OHT producing changes in sea ice, and altering then the MLD around Greenland (Meccia et al., 2022)."

**R2C24**
A more explicit acknowledgment of model limitations—such as the coarse resolution for sea-ice processes and potential biases in AMOC strength—would improve transparency.

Discrepancies between models are discussed in the third paragraph of the "Discussion" section. We have completed this paragraph highlighting the limitations in model resolution and AMOC modelling, as follows: "In line with Sgubin et al. (2017), the analyses performed in this work show that the contribution of each mechanism strongly depends on the model, with some models like MRI-ESM2-0 showing strong alterations of the AMOC, some models like CNRM-ESM2-1 highlighting persistent changes in the sea ice, and some others like IPSL-CM6A-LR and CanESM5 simulating a relevant impact of the SMOC. This model-dependent behavior highlights the current uncertainties for the simulation of the ocean circulation and sea ice responses, which are probably limited by the coarse resolutions considered for these experiments, unable to resolve potentially key contributions from mesoscale eddies and sea ice leads."

**R2C25**
Structuring the discussion into thematic subsections ('Sea ice feedbacks', 'AMOC sensitivity', 'Combined mechanisms') would greatly enhance readability.

We have changed the structure of the discussion by swapping the order of paragraph 3 and 4 and moving the last sentences of paragraph 2 to paragraph 3. In such a way, the discussion is organized as follows:

- Temperature asymmetries
- Possible mechanisms explaining the temperature asymmetries
- Link between mechanisms and temperature asymmetries
- Model discrepancies

**R2C26**
Conclusions

The conclusion should begin by restating the study's objective and summarizing main key findings. It should briefly emphasize the study's novelty, explicitly state inherent limitations (e.g., model uncertainties), and suggest two or three clear avenues for future research, such as high-resolution model validation or observational comparison.

We have reworded the conclusions to follow the suggested structure, as follows: "This work provides a comprehensive analysis of CMIP6 overshoot scenarios, with the goal of identifying the main large-scale temperature patterns present in the post-overshoot climate, understanding inter-model differences, and how ocean circulation and sea ice changes contribute to shape these patterns. These analyses show a link between permanent changes in the OHT, AMOC and sea ice after the overshoot and the emergence of temperature asymmetries in the post-overshoot climate, between NH and SH, between mid- and high-latitude regions of the NH and between eastern and western Southern Ocean. They also show substantial inter-model differences in the relative contributions of meridional overturning circulation and sea ice changes to these asymmetries. This paper sheds light on the processes associated with a variety of model responses to the same forcing, in this case overshoot scenarios. Different processes dominate in different models, and the strength of changes in some specific features such as AMOC or MLD is associated with their model-specific mean state. Such in-depth analysis of changes and associated processes across models is the basis to better understand projected changes and can ultimately help to reduce the uncertainty of expected changes. Additional efforts would be needed to better understand model differences with regard to ocean circulation and sea ice changes, also considering new models at substantially enhanced resolution, while comparisons with observational data, in particular for the MLD and AMOC strength, could help to reduce model uncertainties."

---

## Referee Report (RR1)

**Review 2nd round – "Contribution of meridional overturning circulation and sea ice changes to large-scale temperature asymmetries in CMIP6 overshoot scenarios"**

I thank the authors for their detailed responses to my previous comments and for the substantial work invested in revising the manuscript. The revisions have improved the clarity and structure of the paper, and the added explanations help strengthen the overall narrative. That said, I believe further work is still needed before the manuscript can be accepted. I recommend **moderate revisions**: the manuscript is publishable after clarifying the role of internal variability, slightly expanding the interpretation of MLD–AMOC links, and refining some methodological justifications. Improvement of the text, particularly in the captions of the figures, is also needed, and minor comments are provided.

**Moderate comments**

1. While Fig. 1 and Appendix A help, the rationale for including/excluding certain basins in the main analysis could be explained earlier in the methods section.

2. It would be helpful to justify why Atlantic-only asymmetries are not given more prominence given the strong AMOC focus.

3. The paper notes the role of internal variability (e.g., differences between ensemble members in Figs. 9d and 10h), but a more quantitative assessment would strengthen the argument. For example, could a formal separation of forced vs. internal variability be attempted (e.g., through signal-to-noise ratios)?

4. The analysis shows strong correlations between climatological MLD and AMOC slowdown, but the causal mechanisms could be better explained. Are high MLD models more sensitive to buoyancy forcing, or is this an artefact of model resolution/parameterization? A brief physical interpretation would be useful.

5. The $R^2$ values are low for the ensemble means, yet the text emphasizes a link in some models. It would be useful to present a table summarizing which models exhibit statistically significant relationships and under which scenario.

6. The SMOC–SSW/SSE asymmetry link is weaker (low $R^2$). Could eddy compensation or wind stress changes be confounding factors here? Even a brief discussion of alternative explanations would strengthen the section.

7. The multi-model means give equal weight to each model regardless of ensemble size. While this is reasonable, the authors could comment on whether weighting by number of realizations changes the conclusions.

8. The manuscript is generally well written but could be tightened in places to avoid repetition (e.g., several paragraphs restate that responses are model-dependent and linked to MLD climatology).

**Minor comments**

**General:**

- Sometimes "mid-latitudes" and other times "medium latitudes" are used — standardize to "mid-latitudes."

- Several long sentences would benefit from breaking into two for readability, especially those containing multiple subordinate clauses (e.g., the sentence starting "The analysis of these scenarios shows that even if global temperatures revert…").

- Keep tenses consistent when describing findings from the literature — mix of present ("is characterized") and past ("was associated") could be harmonized.

**Abstract:**

- line 6: "at regional level" → "at the regional level"
- line 7: "the situation post-overshoot may differ from the situation pre-overshoot…" → "... regional conditions post-overshoot may still differ from those pre-overshoot..."
- line 3 and 10  "the sea-ice" → "sea-ice"

**Introdution:**

- In the paragraph  line 53 starting by "However, there are large uncertainties in these changes, with responses of the AMOC to forcing changes that strongly depend on the model considered (Sgubin et al., 2017), please include the papers from Bellomo, 2021 [https://www.nature.com/articles/s41467-021-24015-w](https://www.nature.com/articles/s41467-021-24015-w)

- move Table 1 to after the section titre "Method"

- line 25: "recovering the pre-overshoot temperatures" → remove "the" → "recovering pre-overshoot temperatures".

- line 37: "hemispherical temperature asymmetries" → should be "hemispheric temperature asymmetries"

- line 44: "Relevant hysteresis mechanisms have been found on the large-scale hydrology" → "on" → "in".

- line 54: "a decrease of OHT" → should be "a decrease in OHT". In the the text, I fould 6 other occurences of "a decrease of", please change it to "in" pages 14, 17 (twice), 20, 21, 23

**Method:**

line 89: "Focus" was used in the sentence before. This paragraph is hard to follow. The term "stabilization", "pre overshoot state" or "post overshoot state" have not been defined before, and we get a bit lost with the different periods. Consider adding a simple scheme (timeline) to help visualise.

**Results:**

**Figure 2 and others:** Caption still needs to be a lot reduced. For example, change: "The minimum-to-maximum spread comprised by the individual simulations within each ensemble is included with a shading" by "enveloppes are the min-max values". If EXT and ALL ensembles have been defined in "Method", there is no need to repeat the definition in the caption. Colors for ALL and CNRM model are too similar, change CNRM model to brown for instance. Unclear why we have "CanESM5-EXT" only, and not the extended model simulations. In multi-panel figures (e.g., Fig. 3, Fig. 7), it might help to label subplots with the asymmetry name directly (EN–ES, ENM–ENH, SSW–SSE) for faster reading.

line 148: "Less discrepancies exist" → "Fewer discrepancies exist"

line 160: "Atlantic ocean" → "Atlantic Ocean"

line 173: "has been also identified" → "has also been identified"

---

## Referee Report (RR2)

The manuscript by Roldán-Gómez et al. investigates temperature asymmetries in response to a hypothesized future reduction in greenhouse gases (GHGs). Based on multi-model simulations from CMIP6, the authors show that these asymmetries are primarily driven by sea ice and large-scale ocean circulation processes, particularly the meridional overturning circulation in the Atlantic and Southern Oceans. The Atlantic overturning circulation contributes to hemispheric-scale temperature asymmetries, while the Southern Ocean overturning largely shapes the zonal asymmetry in the Southern Hemisphere. In addition, sea ice changes dominate the temperature response across the middle to high latitudes of the Northern Hemisphere. Despite these robust mechanisms, substantial inter-model discrepancies persist, which may influence the projected temperature response to mitigation efforts. Overall, the findings are interesting and the manuscript is clearly written. I recommend minor revision with several specific comments.

1. Several figures are difficult to interpret and would benefit from revision. For example, Figs. 2a–b, 3, 5, and 6 contain lines that are too thin to be easily distinguished, and the color schemes use shades that are too similar to be clearly identified. I also suggest that the authors consider adjusting the colors of the vertical lines to improve readability.

2. The SSP5-3.4OS ALL scenario generally produces results consistent with those of SSP1-1.9 ALL, although the magnitude differs due to the stronger and earlier $CO_2$ reduction in SSP1-1.9 ALL. Therefore, I recommend that the authors move the SSP1-1.9 figures to the supplementary materials, using them to support the main conclusions drawn from SSP5-3.4OS.

3. The authors use both SSP5-3.4OS EXT and SSP5-3.4OS ALL to explore the temperature response to $CO_2$ reduction. I would suggest the author highlight the differences of the two sets.

---

## Author Response (AR2)

**Responses to reviewer's comments for "Contribution of meridional overturning circulation and sea-ice changes to large-scale temperature asymmetries in CMIP6 overshoot scenarios" (2$^{nd}$ round)**

**Reviewer 1:**

We are grateful to the reviewer for the comments and suggestions, all of which have been helpful for improving the manuscript. We answer to each of the comments below, providing in gray the comments from the review and in black our responses.

I thank the authors for their detailed responses to my previous comments and for the substantial work invested in revising the manuscript. The revisions have improved the clarity and structure of the paper, and the added explanations help strengthen the overall narrative. That said, I believe further work is still needed before the manuscript can be accepted. I recommend moderate revisions: the manuscript is publishable after clarifying the role of internal variability, slightly expanding the interpretation of MLD–AMOC links, and refining some methodological justifications. Improvement of the text, particularly in the captions of the figures, is also needed, and minor comments are provided.

**R1C1**
Moderate comments
1. While Fig. 1 and Appendix A help, the rationale for including/excluding certain basins in the main analysis could be explained earlier in the methods section.

As a general answer, all the analyses based on SST consider all the basins. The suitability of this choice has been verified in the Appendix A by comparing the cross-basin SST with the Atlantic SST. A per-basin analysis is only considered for the OHT and mass transport, for which it is well known that there is a different behavior for Atlantic and Pacific.

This has been clarified in the beginning of the methods section:
- "In a first step, analyses are focused on annual Sea Surface Temperatures (SST), to characterize the temperature asymmetries generated during the overshoot. For that, the temporal evolution of SST and the difference between pre- and post-overshoot states are analyzed, considering all the ocean basins.".
- "The asymmetries have been evaluated for all the basins together, considering the results from Roldán-Gómez et al. (2025) and the existing connections between Atlantic, Southern Ocean and Indo-Pacific basins (Li et al., 2024). To confirm that the results are not sensitive to this choice, results for EN, ES and ENM but including only the Atlantic basin (ENATL, ESATL and ENMATL) are included in Appendix A."

**R1C2**
2. It would be helpful to justify why Atlantic-only asymmetries are not given more prominence given the strong AMOC focus.

Even if the asymmetries are associated with the AMOC, there are mechanisms that redistribute the heat between different basins. This is already discussed in the introduction: "Li et al. (2024) show that the post-overshoot state in SSP5-3.4OS is characterized by a persistent weakening of Atlantic Meridional Overturning Circulation (AMOC), with impacts on the OHT also for the Southern Ocean and, to a lesser extent, for the Indo-Pacific basin."

A clarification has been added in the Appendix A: "Despite this minor difference, the results shown in Fig. A1 and A2 are generally in line with those from Fig. 5 2, 7 and 9, with ENATL-ESATL asymmetries behaving in a similar way as EN-ES asymmetries and ENMATL-ENH asymmetries behaving in a similar way as ENM-ENH asymmetries. This similarity confirms that even if the AMOC is mainly focused on the Atlantic basin, the changes in the OHT impact also the Southern Ocean and the Indo-Pacific basin (Li et al., 2024)."

**R1C3**
3. The paper notes the role of internal variability (e.g., differences between ensemble members in Figs. 9d and 10h), but a more quantitative assessment would strengthen the argument. For example, could a formal separation of forced vs. internal variability be attempted (e.g., through signal-to-noise ratios)?

Even if the results show a contribution of internal variability (particularly for models and experiments providing more simulations, like SSP1-1.9 for CanESM5 and MIROC6), for a complete characterization of internal variability larger ensembles would be needed to be able to isolate the forced signals.

For analysing the role of internal variability in irreversibility processes (and in particular in the link between sea ice and ENM-ENH asymmetries), the optimal selection of experiments may be different from the one of this paper (in particular to consider large ensemble experiments). This would exceed the scope of this paper, focused on analyzing the CMIP6 overshoot scenarios.

**R1C4**
4. The analysis shows strong correlations between climatological MLD and AMOC slowdown, but the causal mechanisms could be better explained. Are high MLD models more sensitive to buoyancy forcing, or is this an artefact of model resolution/parameterization? A brief physical interpretation would be useful.

There is certainly a contribution of model resolution/parameterization, which is already mentioned in the discussion: "This model-dependent behavior highlights the current uncertainties for the simulation of the ocean circulation and sea ice responses, which are probably limited by the coarse resolutions considered for these experiments, unable to resolve potentially key contributions from mesoscale eddies and sea ice leads. The discrepancies between models can be associated with different climatological values for the MLD in the Subpolar North Atlantic region and in the Southern Ocean depending on the model."

However, other factors may also contribute to link climatological MLD and AMOC slowdown. To

identify all the contributing factors a more detailed per-model analysis would be needed.

We have included this in the discussion section, as a future work: "Despite these preliminary results, a more detailed per-model analysis would be needed to fully understand the physical mechanisms explaining this relationship. Identifying such relationships between climatological characteristics and projected changes could enable potential emergent constraints (Hall et al., 2019), if a robust observational estimate of MLD could be obtained."

**R1C5**

5. The $R^2$ values are low for the ensemble means, yet the text emphasizes a link in some models. It would be useful to present a table summarizing which models exhibit statistically significant relationships and under which scenario.

When considering a single model, we have only one point (ensemble average) or a few points (if considering individual simulations) in Fig. 7, 8, 9, 10 and A2. This is not enough to make any statement on the significance of relationships for individual models. What we can say is that certain models show larger values for one of the indices and also larger (or smaller) values for another.

This is done for example in these sentences:
- "The models with the most negative differences in the northward Atlantic OHT, like MRI-ESM2-0 and CNRM-ESM2-1, are also those showing the coldest temperatures in EN with respect to ES"
- "For the case of SSP1-1.9, MRI-ESM2-0 and CNRM-ESM2-1 are also the models showing the strongest OHT reduction and EN-ES asymmetry"
- "Indeed, in models like UKESM1-0-LL, CMCC-ESM2, IPSL-CM6A-LR, CanESM5 and MIROC6 the individual simulations with a larger decline in the NH sea ice area after the overshoot, show the strongest ENM-ENH asymmetry."
- "the models with a larger OHT increase (e.g. UKESM1-0-LL, IPSL-CM6A-LR, and CNRM-ESM2-1) tend to simulate the largest temperature asymmetries"

**R1C6**

6. The SMOC–SSW/SSE asymmetry link is weaker (low $R^2$). Could eddy compensation or wind stress changes be confounding factors here? Even a brief discussion of alternative explanations would strengthen the section.

Yes, indeed. There are other factors that can contribute to temperature asymmetries in the SH. The paragraph has been completed with that: "For the case of SSW-SSE asymmetry, the large spread among the models reduces the R2 coefficient between OHT and temperature differences, to 0.43 for SSP5-3.4OS EXT (Fig. 10e) and to 0.17 for SSP1-1.9 ALL (Fig. 10f). Other factors like eddy compensation or wind stress changes may also have a relevant role in this area, contributing also to reduce the R2 coefficient.".

**R1C7**

7. The multi-model means give equal weight to each model regardless of ensemble size. While this

is reasonable, the authors could comment on whether weighting by number of realizations changes the conclusions.

The overall conclusions are not changed, but from a methodological point of view, weighting per simulation and not per model would give a strong weight to some particular models (with a larger number of simulations). Considering the strong differences across the models, this is not suitable, as this could mask the variability within the other models.

This has been clarified in the methods section: "Considering the disparity in the number of simulations available for each model (only one SSP1-1.9 simulation for CNRM-ESM2-1, FGOALS-g3 and GFDL-ESM4, and up to 50 SSP1-1.9 simulations for CanESM5 and MIROC6), the use of the same weight for each simulation would generate an ensemble average mostly driven by a few models with large ensembles. To avoid that, the ensemble averages of the ALL and EXT ensembles are computed by averaging all the simulations from each model to obtain a per-model average in a first step and by averaging all the models in a second step, so that all the models contribute with the same weight to the multi-model average."

**R1C8**
8. The manuscript is generally well written but could be tightened in places to avoid repetition (e.g., several paragraphs restate that responses are model-dependent and linked to MLD climatology).

To avoid repetition:

- The first paragraph of section 3.2 :

"The results from Fig. 2 and Fig. 3 show persistent changes in the regional temperatures after the overshoot, but these results strongly depend on the model. To better understand the reasons behind this behavior, different mechanisms potentially contributing to regional hysteresis in case of over-shoot are analyzed. Among these mechanisms, the alteration of the meridional overturning circulation plays a major role (Li et al., 2024)."

Has been replaced by:

"The alteration of the meridional overturning circulation plays a major role on the regional hysteresis in case of overshoot (Li et al., 2024)"

- The following sentence has been removed from the last paragraph of the discussion:

"This paper sheds light on the processes associated with a variety of model responses to the same forcing, in this case overshoot scenarios. Different processes dominate in different models, and the strength of changes in some specific features such as AMOC or MLD is associated with their model-specific mean state."

**R1C9**

Minor comments
General:
- Sometimes "mid-latitudes" and other times "medium latitudes" are used — standardize to "mid-latitudes."

All the occurrences of "medium latitudes" have been replaced by "mid-latitudes".

**R1C10**

- Several long sentences would benefit from breaking into two for readability, especially those containing multiple subordinate clauses (e.g., the sentence starting "The analysis of these scenarios shows that even if global temperatures revert…").

The sentence has been divided as: "The analysis of these scenarios shows different behaviors for global and regional climates. Even if global temperatures revert, the impact on regional temperature, precipitation and climate extremes may remain for decades (Pfleiderer et al., 2024)."

The same has been done for the following sentences:
- "The growing probability of exceeding the temperature targets of the Paris Agreement of 2015 (Raftery et al., 2017), as a result of delays in implementing effective mitigation measures (IPCC, 2022), increases the likelihood of overshoot scenarios. In these scenarios, global average temperature surpasses the target of $1.5_\circ$ C above pre-industrial levels (United Nations / Framework Convention on Climate Change, 2015) and is reduced to the target afterwards with net-negative emissions (Gasser et al., 2015)."
- "The analysis of the SSP1-1.9 and SSP5-3.4OS scenarios confirms the relevant role of hysteresis mechanisms in shaping regional temperature and precipitation after global temperature overshoots. Hysteresis is understood as the dependence of the climate system not only on the current $CO_2$ concentration but on the $CO_2$ pathway."

**R1C11**

- Keep tenses consistent when describing findings from the literature — mix of present ("is characterized") and past ("was associated") could be harmonized.

As suggested, the past tenses have been replaced by present tenses:
- "increasing phase is not totally released "
- "Hysteresis in the sea ice coverage is also identified"
- "This regional irreversibility, understood as a post-overshoot state different from the pre-overshoot state with the same $CO_2$ concentration levels and with the same global temperature, is associated with hemispherical temperature"
- "Changes in the sea ice and ocean circulation are identified as potential sources of hysteresis"
- "Relevant hysteresis mechanisms are found on the large-scale hydrology"

**R1C12**

Abstract:
• line 6: "at regional level" → "at the regional level"

The sentence has been modified as suggested in the comment.

**R1C13**

• line 7: "the situation post-overshoot may differ from the situation pre-overshoot…" → "… regional conditions post-overshoot may still differ from those pre-overshoot..."

The sentence has been modified as suggested in the comment.

**R1C14**

• line 3 and 10 "the sea-ice" → "sea-ice"

The two occurrences have been modified as suggested in the comment.

**R1C15**

Introdution:
- In the paragraph line 53 starting by "However, there are large uncertainties in these changes, with responses of the AMOC to forcing changes that strongly depend on the model considered (Sgubin et al., 2017), please include the papers from Bellomo, 2021 https://www.nature.com/articles/s41467-021-24015-w

Reference to Bellomo et al. (2021) has been included.

**R1C16**

- move Table 1 to after the section titre "Method"

Table 1 has been moved after the section title "Method".

**R1C17**

- line 25: "recovering the pre-overshoot temperatures" → remove "the" → "recovering pre-overshoot temperatures".

The sentence has been modified as suggested in the comment.

**R1C18**

- line 37: "hemispherical temperature asymmetries" → should be "hemispheric temperature asymmetries"

The sentence has been modified as suggested in the comment.

**R1C19**

- line 44: "Relevant hysteresis mechanisms have been found on the large-scale hydrology" → "on" → "in".

The sentence has been modified as suggested in the comment.

**R1C20**

- line 54: "a decrease of OHT" → should be "a decrease in OHT". In the the text, I fould 6 other occurences of "a decrease of", please change it to "in" pages 14, 17 (twice), 20, 21, 23

All the occurrences of "a decrease of" have been replaced by "a decrease in". Accordingly, all the occurrences of "a increase of" have been replaced by "a increase in".

**R1C21**

Method:
line 89: "Focus" was used in the sentence before. This paragraph is hard to follow. The term "stabilization", "pre overshoot state" or "post overshoot state" have not been defined before, and we get a bit lost with the different periods. Consider adding a simple scheme (timeline) to help visualise.

The paragraph has been reworded to make it more clear: "In a first step, analyses are focused on annual Sea Surface Temperatures (SST), to characterize the temperature asymmetries generated during the overshoot. For that, the temporal evolution of SST and the difference between pre- and post-overshoot states are analyzed, considering all the ocean basins. To highlight the long-term variability, the temporal evolutions are filtered with a 10 year moving average. The post-overshoot state is defined as a 20-year period from 2220 to 2239 for SSP5-3.4OS EXT and from 2080 to 2099 for the SSP1-1.9 and the SSP5-3.4OS ALL ensemble."

**R1C22**

Results:
Figure 2 and others: Caption still needs to be a lot reduced. For example, change: "The minimum-to-maximum spread comprised by the individual simulations within each ensemble is included with a shading" by "enveloppes are the min-max values". If EXT and ALL ensembles have been defined in "Method", there is no need to repeat the definition in the caption. Colors for ALL and CNRM model are too similar, change CNRM model to brown for instance. Unclear why we have "CanESM5-EXT" only, and not the extended model simulations. In multi-panel figures (e.g., Fig. 3, Fig. 7), it might help to label subplots with the asymmetry name directly (EN–ES, ENM–ENH, SSW–SSE) for faster reading.

As suggested, captions have been modified to:
- Change the sentence explaining the shading: "Envelopes show the min-max values within each ensemble".
- Remove the definition of ALL and EXT ensembles: "including the ALL and EXT ensembles, the ensemble average for each individual model and the individual extended simulations".

Regarding the colors, brown is already used for MIROC6, so it cannot be used for CNRM. As explained in the answer to R2C1, considering the number of models included in the figures, we could not find another color scheme that provides a better readability than the selected one. To

avoid any misunderstanding on the figures, we have increased the width of the lines, so that the colors are clearer.

Regarding the CanESM5-EXT, we have indeed all the extended simulations in the figures. But for MRI-ESM2-0, CNRM-ESM2-1 and IPSL-CM6A-LR there is only one simulation of SSP5-3.4OS, which is the extended one. There is then no need to distinguish between ensemble average and extended simulation, as done for CanESM5. To avoid any misunderstanding, we have clarified this in the methods section: "For MRI-ESM2-0, CNRM-ESM2-1 and IPSL-CM6A-LR the only available SSP5-3.4OS simulation covers the extended period, so there is no difference between the extended simulation and the ensemble average. However, for CanESM5 there is one simulation covering the extended period and 4 simulations covering only until 2100. In that case, the extended simulation is identified as CanESM5-EXT, while the average of the 5 simulations is identified as CanESM5."

Regarding the label of subplots, we have kept the (a), (b), ... labeling (following journal recommendations), but we have included the asymmetry name when citing them in the text:
- "gold line of SST EN-ES in Fig. 3a"
- "gold line of SST ENM-ENH in Fig. 3b"
- "purple line of SST EN-ES in Fig. 3a,b"
- "gray line of SST ENM-ENH in Fig. 3c"
- "red line of SST ENM-ENH in Fig. 3d"
- "gold and purple lines of SST ENM-ENH in Fig. 3c,d"
- "dark orange line of SST ENM-ENH in Fig. 3c"
- "orange line of SST ENM-ENH in Fig. 3d"
- "turquoise line of SST SSW-SSE in Fig. 3f"

**R1C23**
line 148: "Less discrepancies exist" → "Fewer discrepancies exist"

The sentence has been modified as suggested in the comment.

**R1C24**
line 160: "Atlantic ocean" → "Atlantic Ocean"

The sentence has been modified as suggested in the comment.

**R1C25**
line 173: "has been also identified" → "has also been identified"

Considering the comment R1C11, the sentence has been changed to present tense.

**Reviewer 2:**

We are grateful to the reviewer for the comments and suggestions, all of which have been helpful for improving the manuscript. We answer to each of the comments below, providing in gray the comments from the review and in black our responses.

The manuscript by Roldán-Gómez et al. investigates temperature asymmetries in response to a hypothesized future reduction in greenhouse gases (GHGs). Based on multi-model simulations from CMIP6, the authors show that these asymmetries are primarily driven by sea ice and large-scale ocean circulation processes, particularly the meridional overturning circulation in the Atlantic and Southern Oceans. The Atlantic overturning circulation contributes to hemispheric-scale temperature asymmetries, while the Southern Ocean overturning largely shapes the zonal asymmetry in the Southern Hemisphere. In addition, sea ice changes dominate the temperature response across the middle to high latitudes of the Northern Hemisphere. Despite these robust mechanisms, substantial inter-model discrepancies persist, which may influence the projected temperature response to mitigation efforts. Overall, the findings are interesting and the manuscript is clearly written. I recommend minor revision with several specific comments.

**R2C1**
1. Several figures are difficult to interpret and would benefit from revision. For example, Figs. 2a–b, 3, 5, and 6 contain lines that are too thin to be easily distinguished, and the color schemes use shades that are too similar to be clearly identified. I also suggest that the authors consider adjusting the colors of the vertical lines to improve readability.

As suggested, Fig. 2a-b, 3, 5 and 6 have been modified, to:
- Increase the width of all the lines.
- Use black color for the vertical lines with the year before the overshoot with the same $CO_2$ concentration (dashed line) and global surface air temperature (solid line) as at the end of the run.
- Use black color for the horizontal lines with the value of the variable for the years with the same $CO_2$ concentration (dashed line) and global surface air temperature (solid line) as at the end of the run.
Considering the number of models included in the figures, we could not find another color scheme that provides a better readability than the selected one.

The references in the text have been adapted accordingly.

The new figures are as follows:

Figure 2:

[Figure]

Figure 3:

[Figure]

Figure 5:

[Figure]

Figure 6:

[Figure]

Figure A1:

[Figure]

2. The SSP5-3.4OS ALL scenario generally produces results consistent with those of SSP1-1.9 ALL, although the magnitude differs due to the stronger and earlier $CO_2$ reduction in SSP1-1.9 ALL. Therefore, I recommend that the authors move the SSP1-1.9 figures to the supplementary materials, using them to support the main conclusions drawn from SSP5-3.4OS.

As discussed in the answers to comments R1C1 and R1C5 from the first round of review, even if the results may be similar to those of SSP5-3.4OS, we still consider that the results for SSP1-1.9 are relevant, since this experiment includes more models and simulations than SSP5-3.4OS, and allows for a more complete assessment of model discrepancies. In addition, it also allows for assessing the role of internal variability, since it contains large ensembles of simulations, like those from CanESM5, MIROC6 and MPI-ESM1-2-LR. For this reason, we prefer to keep them in the main text.

**R2C3**

3. The authors use both SSP5-3.4OS EXT and SSP5-3.4OS ALL to explore the temperature response to CO2 reduction. I would suggest the author highlight the differences of the two sets.

The differences between EXT and ALL ensembles are described in the methods section. To make it

more clear, we have modified the paragraph to clearly reflect the number of simulations, the covered period and the goal of each ensemble: "The ALL ensemble of SSP5-3.4OS contains 16 simulations from 8 models, covering the period from from 1850 to 2100, while the EXT ensemble of SSP5-3.4OS contains only 4 simulations from 4 models, but covering an extended period from 1850 to 2300. Both ensembles are then complementary, being the ALL ensemble mostly used to analyze the inter-model and the EXT ensemble mostly used to analyze the long-term stabilization after the overshoot."